# Specialized shuttle proteins recognize Type IX secretion signals and target effectors to their final destinations in *Flavobacterium johnsoniae*

Maelle Paillat [1], Caterina Comas Hervada[1], Stéphane Audebert [2], Eric Cascales [1] & Thierry Doan [1] ✉

Members of the phylum *Bacteroidota* utilize the type IX secretion system (T9SS) to transport diverse substrates into the environment or onto their surface. T9SS substrates feature a Sec-dependent signal peptide for export to the periplasm and a conserved C-terminal domain (CTD), recognized by the T9SS, for translocation across the outer membrane. Following translocation, substrates engage with a shuttle protein, which ensures their final localization. Most CTDs are classified into two major families. Type A CTDs are all recognized by the PorV shuttle. Recognition and transport of Type B CTDs remain less explored. *Flavobacterium johnsoniae* encodes 12 Type B substrates, often genetically linked to genes encoding PorP/SprF-like shuttle proteins. We demonstrate that two Type B substrates indeed rely on their cognate PorP/SprF specialized shuttle proteins for secretion and identify the shuttle responsible for the secretion of three orphan Type B CTDs. Our findings also reveal that five conserved motifs within Type B CTDs are necessary for secretion but not sufficient for their specific recognition by cognate shuttle proteins. Our results further suggest that CTDs contain a secretion signal, sufficient for secretion of substrates by the T9SS, and a targeting signal, which directs substrates to their final localization.

In bacteria, protein secretion plays a crucial role in a wide range of physiological processes, including nutrient acquisition, intercellular communication, colonization of hosts or of environmental niches, and pathogenesis. The Type IX secretion system (T9SS) is a specialized secretion apparatus exclusively found in bacteria of the phylum *Bacteroidota*[1,2].

The T9SS is a multiprotein nanomachine comprised of 15 to over 20 proteins that spans the entire cell envelope. Recent structural advances have shed light on the architecture of this machine. Cryo-electron tomography analyses of *Porphyromonas gingivalis* cells revealed that the T9SS organizes as a caged-ring structure in this bacterium[3]. A large, outer membrane-associated, ring-like structure composed of PorK (GldK in *F. johnsoniae*) and PorN (GldN) is connected to the inner membrane by 18 pillars[4–6]. The pillars are comprised of a PorM (GldM) dimer associated with a pentamer of PorL (GldL) embedded in the inner membrane. PorL$_5$M$_2$ complexes constitute proton-motive force-dependent molecular motors that energize

effector secretion and, in *F. johnsoniae*, gliding motility[6–10]. Eight outer membrane-spanning translocons localize within the PorKN ring. These translocons consist of a large beta barrel, Sov (SprA), with a lateral opening that allows an associated shuttle protein to screen and pull out substrates[11–14]. Finally, the attachment complex, which includes PorQ, PorZ, and the PorU sortase, processes substrates by cleaving the T9SS-signal sequence (described below) and anchoring them to anionic lipopolysaccharides (A-LPS)[15–20].

The T9SS supports the secretion of a large repertoire of effectors, varying in size and function[21]. It was first identified in the oral pathogen *P. gingivalis* and the soil bacterium *F. johnsoniae*. In *P. gingivalis*, the T9SS is responsible for the secretion of virulence factors called gingipains, which contribute to periodontal tissue degradation[22–26]. In motile bacteria such as *F. johnsoniae*, the T9SS notably supports the secretion of adhesins involved in gliding motility[27]. After being secreted by the T9SS, the major gliding adhesin SprB remains tethered to the surface and moves rapidly along a

[1]Laboratoire d'Ingénierie des Systèmes Macromoléculaires, Institut de Microbiologie de la Méditerranée, Aix-Marseille Université, CNRS, UMR7255, Marseille, France. [2]Inserm U1068, CNRS, Institut Paoli-Calmettes, Centre de Recherche en Cancérologie de Marseille (CRCM), Marseille Protéomique, Aix-Marseille Université, Marseille, France. ✉e-mail: tdoan@imm.cnrs.fr

closed helical path[28]. All T9SS effectors possess an N-terminal Sec-dependent signal peptide, which supports their export to the periplasm, where they are recruited by the T9SS. T9SS substrates recognition and sorting is mediated by a conserved C-terminal domain (CTD) that serves as a secretion signal[29], ranging from approximately 100 amino acids for Type A CTDs to 150-235 amino acids for Type B CTDs. The CTD is necessary and sufficient for secretion, as heterologous proteins fused to a CTD are secreted by the T9SS[21,30,31]. Different families of CTDs have been identified based on their sequence, suggesting that the T9SS can recognize multiples secretion signals[32]. The Type A CTD (TIGR04183) is the most common and the best-characterized T9SS sorting signal[29–31,33,34]. Previous studies showed that substrates carrying a Type A CTD require PorV as a shuttle protein to be collected from the SprA translocons[11,13,35]. Structures of several Type A CTDs have been solved: Type A CTDs are comprised of seven antiparallel β-strands that pack as a β-sandwich Ig-like fold[18,36–38]. Recent structural analysis and cryo-EM data provided valuable insights into the interaction between PorV and Type A CTDs, identifying key motifs and residues involved in this interaction[11,13,39]. By contrast, substrates bearing Type B CTDs (TIGR04131) are less abundant or even absent in some T9SS-encoding genomes[21,31]. Type B CTDs differ in both sequence and structure from Type A CTDs[21,32]. Previous studies suggested that the minimal sequence required for efficient secretion of Type B CTDs is longer than that of Type A CTDs (up to 235 amino acids), extending beyond the conserved C-terminal domain[21]. No Type B CTD structure has been solved to date; however, structure predictions suggest that it comprises a four-strand β-sheet facing a less ordered region (Supplementary Fig. 1A). Unlike Type A substrates, the secretion of Type B substrates does not depend on PorV[35]. Instead, for two substrates bearing a Type B CTD in *F. johnsoniae*, it has been shown that secretion requires a cognate PorP/SprF-like shuttle protein. In both these cases, the gene encoding the cognate SprF-like protein is localized directly downstream of the Type B substrate-encoding gene[21]. This genetic organization is likely to be conserved, as a *porP/sprF*-like gene is commonly found directly downstream of genes encoding a Type B CTD-containing protein. These findings suggest that the secretion of each Type B substrate relies on its specific shuttle protein. However, in some cases, "orphan" Type B substrates are not genetically linked to *sprF*-like genes, and hence their mechanism of extrusion from the translocon remains to be understood[21].

By using pull-down coupled with mass spectrometry analyses and functional assays, we show here that the three "orphan" Type B CTD-containing substrates encoded within the *F. johnsoniae* genome use the same shuttle protein. Through mutagenesis of Type B CTDs, we then demonstrate that conserved motifs in Type B CTDs are necessary but not sufficient for the specific interaction between each substrate and its cognate shuttle. Finally, by characterizing the localization and behavior of a fluorescent reporter fused to these CTDs, our data support a model in which Type B CTDs serve both as a secretion signal via the T9SS and a targeting signal to direct substrates to their final destination.

## Results

### Secretion of T9SS Type B substrates requires specific shuttle proteins

Previous work used sfGFP fusion to the C-terminal domain (CTD) of T9SS effectors to monitor their secretion and to identify the minimal sequence required for recognition and transport[21,31]. As shown previously[31], sfGFP is efficiently secreted once fused upstream of the Type A CTD of RemA or of AmyB, even when highly produced from a replicative plasmid (Fig. 1A, B). By contrast, secretion of sfGFP fused to the Type B CTD of SprB or of Fjoh_3952 is only observed when the cognate shuttle protein, SprF or Fjoh_3951, respectively, encoded by the immediate downstream gene, is co-expressed from the same plasmid (Fig. 1A, B)[21]. Considering the genetic organization of the genes encoding Type B substrates and PorP/SprF-like shuttle proteins, it was hypothesized that the amount of endogenous PorP/SprF proteins produced is insufficient to ensure the efficient secretion of

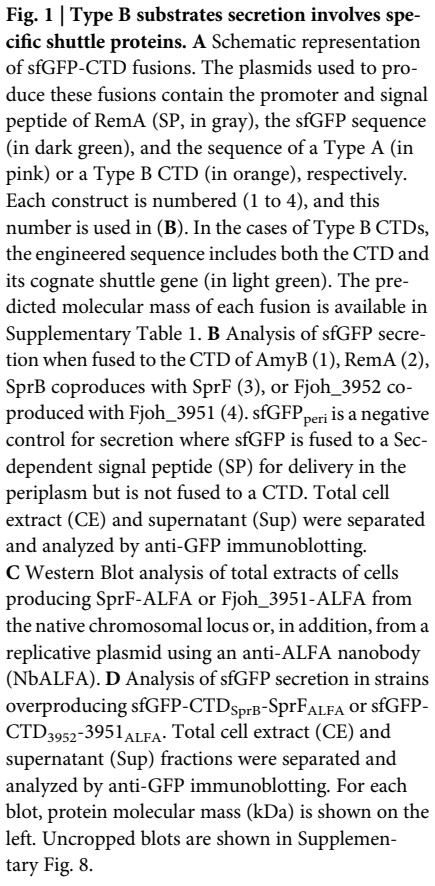

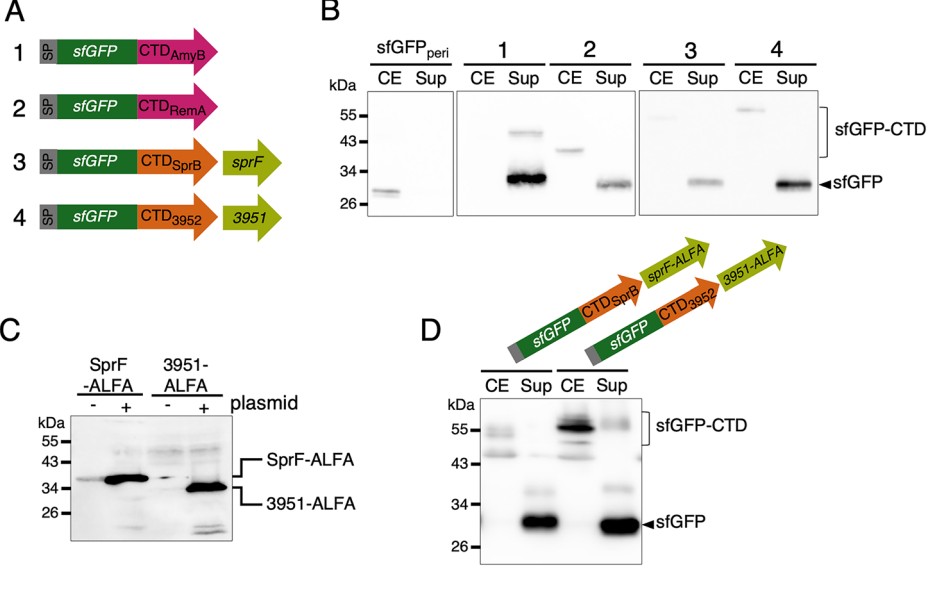

**Fig. 1 | Type B substrates secretion involves specific shuttle proteins. A** Schematic representation of sfGFP-CTD fusions. The plasmids used to produce these fusions contain the promoter and signal peptide of RemA (SP, in gray), the sfGFP sequence (in dark green), and the sequence of a Type A (in pink) or a Type B CTD (in orange), respectively. Each construct is numbered (1 to 4), and this number is used in (**B**). In the cases of Type B CTDs, the engineered sequence includes both the CTD and its cognate shuttle gene (in light green). The predicted molecular mass of each fusion is available in Supplementary Table 1. **B** Analysis of sfGFP secretion when fused to the CTD of AmyB (1), RemA (2), SprB coproduces with SprF (3), or Fjoh_3952 coproduced with Fjoh_3951 (4). sfGFP_peri is a negative control for secretion where sfGFP is fused to a Sec-dependent signal peptide (SP) for delivery in the periplasm but is not fused to a CTD. Total cell extract (CE) and supernatant (Sup) were separated and analyzed by anti-GFP immunoblotting. **C** Western Blot analysis of total extracts of cells producing SprF-ALFA or Fjoh_3951-ALFA from the native chromosomal locus or, in addition, from a replicative plasmid using an anti-ALFA nanobody (NbALFA). **D** Analysis of sfGFP secretion in strains overproducing sfGFP-CTD_SprB-SprF_ALFA or sfGFP-CTD_3952-3951_ALFA. Total cell extract (CE) and supernatant (Sup) fractions were separated and analyzed by anti-GFP immunoblotting. For each blot, protein molecular mass (kDa) is shown on the left. Uncropped blots are shown in Supplementary Fig. 8.

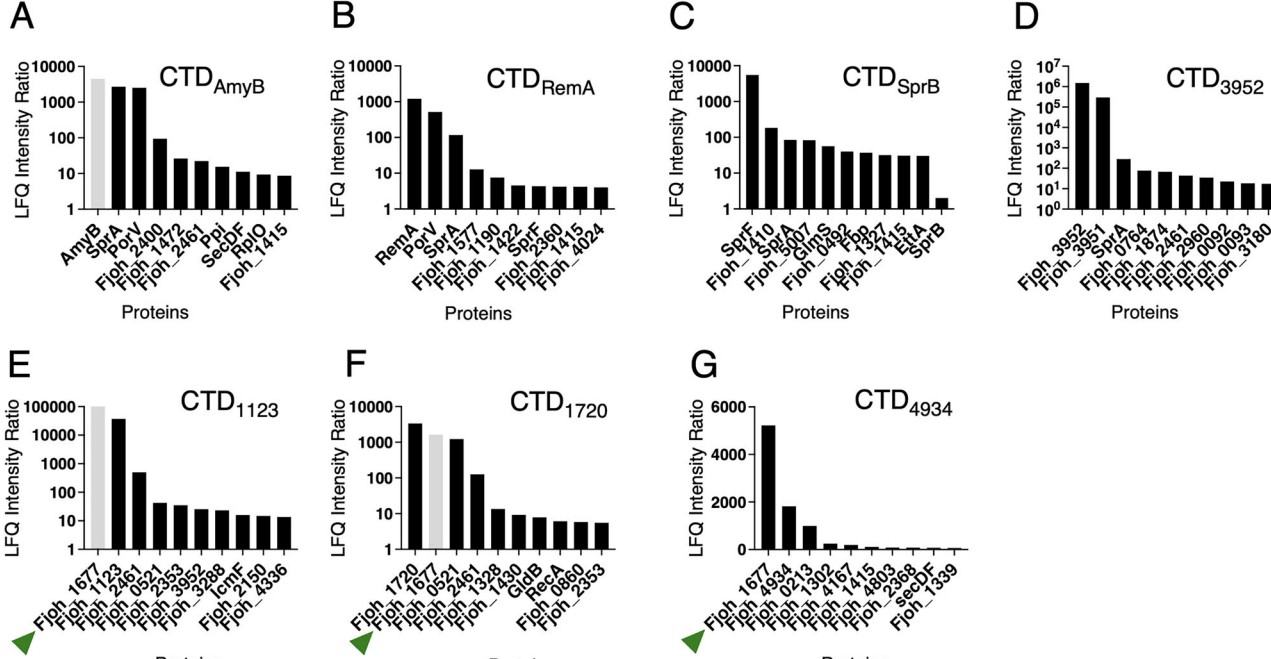

**Fig. 2 | Identification of shuttle proteins associated to different CTDs.** *F. johnsoniae* strains expressing sfGFP-CTD$_{AmyB}$ (**A**), sfGFP-CTD$_{RemA}$ (**B**), sfGFP-CTD$_{SprB}$-SprF (**C**), sfGFP-CTD$_{Fjoh\_3952}$-Fjoh_3951 (**D**), sfGFP-CTD$_{Fjoh\_1123}$ (**E**), sfGFP-CTD$_{Fjoh\_1720}$ (**F**), or sfGFP-CTD$_{Fjoh\_4934}$ (**G**) were subjected to crosslink followed by co-immunoprecipitation using anti-GFP agarose beads. The strain expressing sfGFP without a fused CTD (sfGFP$_{peri}$) was used as control. The ratio of abundance of enriched proteins (LFQ intensity ratio) for each strain expressing a sfGFP-CTD fusion relative to the strain producing sfGFP$_{peri}$ were plotted. When the protein was absent from the sfGFP$_{peri}$ strain, an arbitrary value was assigned to allow ratio calculation, and the corresponding bar was colored gray. The green triangles highlight the same PorP/SprF shuttle associated with the three orphan Type B substrates. For each strain, analysis was done on three biological replicates, each injected three times on mass spectrometers. Data are the mean of three independent replicates.

both the native Type B substrate and the corresponding highly produced sfGFP-CTD fusion[21]. To determine if this hypothesis is correct, we generated SprF-ALFA and Fjoh_3951-ALFA fusions. The ALFA tag is a 13-amino-acid peptide that is specifically recognized by the NbALFA nanobody with high affinity[40]. We then performed Western blot analysis of shuttle protein levels expressed either only from the endogenous locus or from the endogenous locus and the replicative plasmid used for overexpression. While overproduction of the tagged shuttle proteins from a replicative plasmid was easily detected, protein production from the endogenous locus was barely visible in the case of SprF-ALFA and not detected in the case of Fjoh_3951-ALFA (Fig. 1C). Furthermore, these ALFA fusions are functional because they support the secretion of their overproduced cognate substrates, sfGFP-CTD$_{SprB}$ or sfGFP-CTD$_{Fjoh\_3952}$ respectively (Fig. 1D). Hence, we conclude that for Type B CTDs, the low abundance of the shuttle proteins in the wild-type strain explains the defective secretion of overproduced substrates. In addition, these results imply that the Type A-dependent PorV shuttle protein is likely produced in excess or recycled, ensuring handling of all Type A CTD substrates, including highly produced sfGFP-CTD fusions. By contrast, Type B CTD substrates and cognate shuttles should be produced in stoichiometric amount, further suggesting a specific function of the shuttle after secretion, which cannot be recycled to support the secretion of other substrates.

## The three orphan Type B CTD substrates, Fjoh_1123, Fjoh_1720 and Fjoh_4934, share the same shuttle protein, Fjoh_1677

Among the 12 genes encoding Type B CTD-containing proteins predicted in *F. johnsoniae*, only 9 are physically adjacent to a gene encoding a SprF-like protein[21]. This observation raised the question as to how the three remaining orphan Type B CTD substrates, Fjoh_1123, Fjoh_1720, and Fjoh_4934, are collected from the SprA translocon. To identify the shuttle(s) protein(s) handling these three orphan Type B substrates, we performed co-immunoprecipitation experiments. Detergent-solubilized cell extracts of

chemically cross-linked cells producing sfGFP fused to different CTDs were immobilized on anti-GFP-conjugated agarose, and enriched proteins were subjected to label-free quantitative (LFQ) mass spectrometry analyses (see Experimental Procedures). Controls included sfGFP fusions to the previously characterized AmyB and RemA Type A CTDs, and SprB and Fjoh_3952 Type B CTDs co-produced with their cognate shuttle protein, and the periplasmic sfGFP (sfGFP$_{peri}$) alone. The four sfGFP-CTD fusions were efficiently secreted, while the periplasmic version of the sfGFP was detected only in the cell extract fraction (Fig. 1B). The ratio of abundance of co-immunoprecipitated proteins from cells producing sfGFP-CTD fusions relative to cells producing sfGFP$_{peri}$ was calculated (LFQ intensity ratios), allowing the identification of proteins specifically enriched in the samples containing sfGFP-CTD fusions (Fig. 2A–G, bars colored in gray correspond to proteins absent from control strain immunoprecipitation [Supplementary Table 2], to which an arbitrary value was assigned to calculate a ratio). As expected, PorV was among the most abundant proteins immunoprecipitated with the Type A CTDs of AmyB (sfGFP-CTD$_{AmyB}$, Fig. 2A) and RemA (sfGFP-CTD$_{RemA}$, Fig. 2B), in agreement with previous results demonstrating that Type A CTDs rely on PorV as shuttle[11,35]. In the case of the SprB (sfGFP-CTD$_{SprB}$, Fig. 2C) and Fjoh_3952 (sfGFP-CTD$_{Fjoh\_3952}$, Fig. 2D) CTDs, their co-produced shuttles, SprF and Fjoh_3951, respectively, were amongst the highest ratios, supporting the idea of a direct and specific interaction between each Type B CTD and its cognate shuttle, as shown previously[21,41]. However, we noticed that SprB was not among the highest LFQ intensity ratios (Fig. 2C). This is because SprB is a very large protein that was detected in significant amount in the control experiment. In the cases of the Fjoh_1123 (Fig. 2E), Fjoh_1720 (Fig. 2F), and Fjoh_4934 (Fig. 2G) CTDs fused to sfGFP, the most enriched protein was Fjoh_1677, an orphan SprF-like protein, whose gene is not encoded directly downstream of a Type B substrate-encoding gene. These results suggest that Fjoh_1677 is the shuttle protein supporting the secretion of the three orphan T9SS Type B substrates Fjoh_1123, Fjoh_1720, and Fjoh_4934, raising the

**Fig. 3 | Coexpression of Fjoh_1677 facilitates the secretion of sfGFP fused to orphan CTDs.**
**A** Schematic representation of the sfGFP-CTD fusions tested. The plasmids used to express these fusions contain the promoter and signal peptide of RemA (SP, in gray), the sfGFP sequence (in dark green), and the sequence of an orphan Type B CTD (in orange). In constructs 6, 8, and 10, the engineered sequences also include the gene encoding the Fjoh_1677 shuttle protein (in light green). The predicted molecular weight of each fusion is available in Supplementary Table 1. **B** Secretion of sfGFP when fused to the CTDs of Fjoh_1123, Fjoh_1720, and Fjoh_4934 was monitored, in the absence (lane 5, 7, and 9, respectively) or presence of Fjoh_1677 (lane 6, 8, and 10, respectively). Total cell extract (CE) and supernatant (Sup) were separated and analyzed by anti-GFP immunoblotting. Protein molecular mass (kDa) is shown on the left. Uncropped blots are shown in Supplementary Fig. 8.

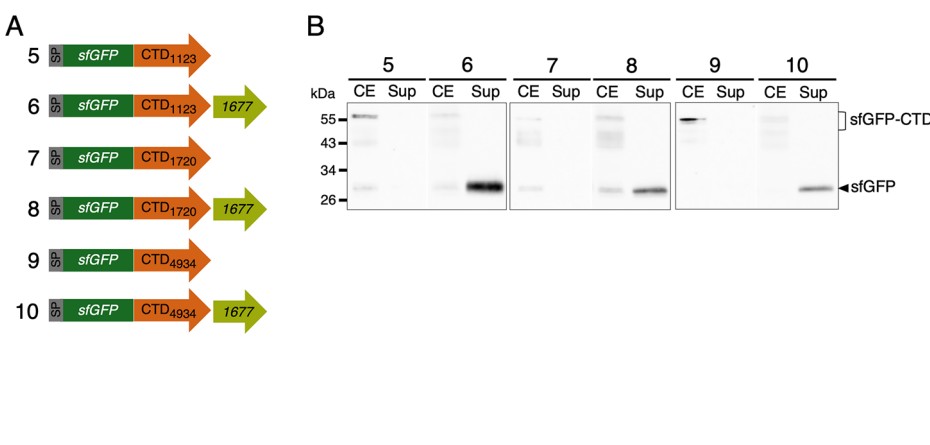

hypotheses that a single shuttle may recognize different Type B CTDs and that crosstalk may exist between other Type B substrate/shuttle protein pairs.

Interestingly, AlphaFold3 complex predictions were consistent with the cognate CTD/shuttle pairs identified by co-immunoprecipitations. First, all cognate pairs were predicted with high confidence with iPTM scores above 0.8 (Supplementary Fig. 2A, B). Second, when multiple CTDs and multiple shuttle sequences were provided (up to 5), AlphaFold3 provided the exact same pairs we identified using biochemistry (Supplementary Fig. 2C). Finally, when multiple shuttle sequences were provided, Alpha-Fold3 always associated any of the three "orphans" Type B CTD substrates Fjoh_1123, Fjoh_1720 and Fjoh_4934, with the Fjoh_1677 shuttle protein, which is consistent with our co-immunoprecipitation results.

## Co-production of Fjoh_1677 facilitates secretion of sfGFP fused to the three orphan Type B CTDs

As previously shown for the CTDs of SprB and Fjoh_3952[21], highly produced sfGFP fusions to the CTDs of the three orphan substrates Fjoh_1123, Fjoh_1720, and Fjoh_4934 cannot be detected by Western blot in the supernatant fraction (Fig. 3). However, Western blot analysis of cell extracts and supernatant fractions of cells co-producing each of these sfGFP-CTD fusions with Fjoh_1677 demonstrated that sfGFP was efficiently secreted (Fig. 3). Based on the results of co-immunoprecipitation and secretion assays, we conclude that Fjoh_1677 is a SprF-like shuttle protein that specifically handles the three orphan Type B substrates Fjoh_1123, Fjoh_1720, and Fjoh_4934.

## Specificity and crosstalk between other Type B CTD/shuttle protein pairs

Sequence alignments revealed that the three orphan CTDs share a significant level of identity (>44%) (Table 1, highlighted in green), which could explain their recognition by the same shuttle protein. Comparisons of Type B CTD sequences to each other also revealed that most CTDs exhibit only low sequence identity (between 20% and 30%). However, a few Type B CTDs share more than 50% of sequence identity (Table 1, highlighted in purple). For example, the CTDs of Fjoh_3952 and Fjoh_1645 share 68% identity. This raises the possibility of crosstalk between CTDs and shuttles from different cognate pairs. To gain insight on potential crosstalk, we investigated whether the CTD of Fjoh_3952 can be handled by the shuttles of three other Type B substrates with which it shares different levels of identity (Fjoh_1645 [68%], Fjoh_3971 [43%], and SprB [26%], shown in bold red) (Table 1). Interestingly, structural alignment of the AlphaFold3 models of these CTDs with the AlphaFold3 model of Fjoh_3952 CTD revealed that the predicted structures of Fjoh_3952 and Fjoh_1645 CTDs

align perfectly (Supplementary Fig. 3A, D). In contrast, some loops are arranged differently between Fjoh_3952 and either SprB or Fjoh_3971 CTDs (Supplementary Fig. 3B–D). The sfGFP-CTD$_{Fjoh\_3952}$ fusion was co-produced with its cognate shuttle, Fjoh_3951, or with the shuttle proteins handling Fjoh_1645, Fjoh_3971, or SprB (Fjoh_1646, Fjoh_3972, or SprF, respectively). Western blot analysis showed that sfGFP-CTD$_{Fjoh\_3952}$ was efficiently secreted when co-produced with Fjoh_3951, but also with the non-cognate Fjoh_1646 shuttle (Fig. 4). By contrast, co-production with Fjoh_3972 or SprF did not promote secretion of sfGFP-CTD$_{Fjoh\_3952}$ into the supernatant (Fig. 4). These results suggest that shuttle proteins can collect and handle non-cognate Type B substrates if they share significant homology with the shuttle-specific substrate.

## Type B CTD conserved motifs are necessary for secretion but not sufficient for recognition by the cognate shuttle protein

Since type B CTD substrates and SprF-like shuttles form cognate pairs, with the three orphans Type B substrates paired with the same orphan shuttle, we hypothesized that they should (1) share conserved molecular signatures to be recognized by SprF-like proteins, and (2) display differences in these motifs to specify the cognate shuttle. Based on sequence alignments, we identified five conserved motifs, named A, B, C, D, and E, within the last 100 amino acids of Type B CTDs (Fig. 5A).

We first asked whether each of the conserved motifs was required for secretion. On the AlphaFold3 model prediction of the CTD of SprB, we observed that motifs B, D, and E are carried by β-strands, while motifs A and C are comprised in loops (Supplementary Fig. 4A). For each motif, we introduced specific mutations altering amino-acid properties without altering the overall model structure (Supplementary Fig. 4B, C). We then tested the effect of each motif mutation on secretion (Fig. 5B). Each mutant was produced but not secreted by the T9SS, indicating that the conserved A, B, C, D, and E motifs of the SprB CTD are all necessary for secretion or recognition by the cognate shuttle SprF.

Next, we asked whether all five conserved motifs are sufficient to support the specificity between a CTD and a shuttle protein. We chose SprB and Fjoh_3952 CTDs to address this question because they rely on different shuttle proteins, SprF and Fjoh_3951, respectively. We swapped all five motifs in the CTD of SprB by the five motifs found in Fjoh_3952, generating CTD$_{SprB-swap}$. Interestingly, AlphaFold3 predictions showed that swapping all five motifs in SprB CTD with those of Fjoh_3952 lead to preferential interaction with Fjoh_3951 instead of SprF (Supplementary Fig. 4D). However, secretion assays showed that this construct was not secreted, whether co-produced with SprF or Fjoh_3951 (Fig. 5C). These results therefore suggest that motifs A to E are necessary for secretion but not sufficient to achieve specific recognition by a shuttle protein.

**Table 1 | Percentage identity matrix of the alignment of the last 100 amino acids of the 12 Type B CTDs found in *F. johnsoniae***

| | Fjoh_4750 | Fjoh_2273 | Fjoh_1985 | Fjoh_4538 | Fjoh_1645 | Fjoh_3952 | Fjoh_3478 | Fjoh_3971 | SprB | Fjoh_1720 | Fjoh_4934 | Fjoh_1123 |
|---|---|---|---|---|---|---|---|---|---|---|---|---|
| **Fjoh_4750** | 100.00 | | | | | | | | | | | |
| **Fjoh_2273** | 25.00 | 100.00 | | | | | | | | | | |
| **Fjoh_1985** | 23.60 | 29.47 | 100.00 | | | | | | | | | |
| **Fjoh_4538** | 26.51 | 37.50 | 48.35 | 100.00 | | | | | | | | |
| **Fjoh_1645** | 23.33 | 32.98 | 37.50 | 62.37 | 100.00 | | | | | | | |
| **Fjoh_3952** | 26.67 | 30.85 | 38.54 | 58.06 | **68.00** | 100.00 | | | | | | |
| **Fjoh_3478** | 29.35 | 38.14 | 38.30 | 41.86 | 34.41 | 32.26 | 100.00 | | | | | |
| **Fjoh_3971** | 32.61 | 43.01 | 46.67 | 53.09 | **43.18** | 43.18 | 57.45 | 100.00 | | | | |
| **SprB** | 28.42 | 33.33 | 27.59 | 30.86 | **26.14** | 26.14 | 34.83 | 37.08 | 100.00 | | | |
| **Fjoh_1720** | 23.16 | 34.07 | 29.21 | 27.50 | 26.44 | 22.99 | 32.61 | 25.00 | 36.56 | 100.00 | | |
| **Fjoh_4934** | 27.37 | 36.26 | 28.41 | 30.38 | 23.26 | 22.09 | 34.78 | 26.04 | 40.86 | 44.44 | 100.00 | |
| **Fjoh_1123** | 23.40 | 31.11 | 30.68 | 29.11 | 22.09 | 23.26 | 31.87 | 27.08 | 41.94 | 45.45 | 56.57 | 100.00 |

Alignments were performed using ClustalOmega[56]. The percentage identity between the three orphan CTDs is highlighted in green. Identity scores above 50% are highlighted in purple, and the scores of the CTDs tested for crosstalk are shown in bold red.

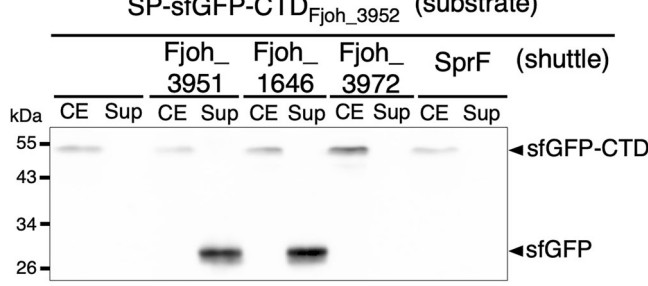

**Fig. 4 | Coexpression of Fjoh_3951 or Fjoh_1646 facilitates the secretion of sfGFP fused to the CTD of Fjoh_3952.** Secretion of sfGFP when fused to the CTD of Fjoh_3952 was monitored when expressed alone, or co-expressed with Fjoh_3951, Fjoh_1646, Fjoh_3972, or SprF. Total cell extract (CE) and supernatant (Sup) were separated and analyzed by anti-GFP immunoblotting. Protein molecular mass (kDa) is shown on the left. Uncropped blots are shown in Supplementary Fig. 8.

## Specific Type B CTD/shuttle pairs specify putative adhesins localization after secretion

Contrarily to substrates with a Type A CTD, which all rely on the same shuttle protein PorV, each Type B CTD-containing substrate relies on a cognate shuttle or, in the case of crosstalk, on a few shuttles of the PorP/SprF family. This raises the idea that each shuttle serves not only to collect the substrate at the exit of the translocon but also to transfer it to its final localization after secretion. It is noteworthy that Type B substrates are known or predicted to be adhesins[21]. Hence, we hypothesize that CTDs and

their cognate shuttle proteins direct adhesin substrates to different functional locations and/or confer different dynamic behaviors.

We first performed time-lapse fluorescence microscopy experiments to track three T9SS model substrates: SprB, an adhesin with a Type B CTD known to be targeted to the gliding machinery[28,42], RemA, a Type A CTD adhesin containing a lectin domain also involved in gliding motility[43], and AmyB, an amylase with a Type A CTD. In order to track these proteins at the single-molecule level, a HaloTag was inserted upstream of the CTD sequence at the native locus (Fig. 6A). SprB-HaloTag-CTD$_{SprB}$ moves along the cell surface following a helical track (Fig. 6B, *left panel*, Supplementary Movie 1[44]), as previously described[28,42]. By contrast, RemA-HaloTag-CTD$_{RemA}$ displayed two different behaviors in our experimental conditions. The first subpopulation consisted of immobile foci (observed in 55% of cells presenting fluorescent signal, $n = 86$) (Fig. 6C, *left panel*, Supplementary Movie 2[44]). These foci could disappear over time (Supplementary Fig. 5), suggesting that RemA is initially anchored to the cell surface before being processed and released into the environment. Consistent with this idea, RemA-HaloTag is detected is the spent medium (Fig. 6E). The second subpopulation corresponds to highly mobile RemA-HaloTag foci (observed in 86% of the cells presenting fluorescent signal, $n = 86$) (Fig. 6C, *middle panel*, Supplementary Movie 3[44]), with a behavior different from that of the SprB adhesin. It is noteworthy that we did not detect any helical behavior of RemA-HaloTag, although RemA was found to move helically in a previous study[43]. This may be explained by the use of different tags, inserted at different positions, to track the protein. Finally, no signal was detected for the AmyB-HaloTag-CTD$_{AmyB}$ fusion (Fig. 6D, *left panel*), suggesting that AmyB is secreted in the medium. However, AmyB secretion was not detected in a dot blot assay and was barely detected in the total cell extract (Fig. 6E), demonstrating that this fusion is produced at very low levels.

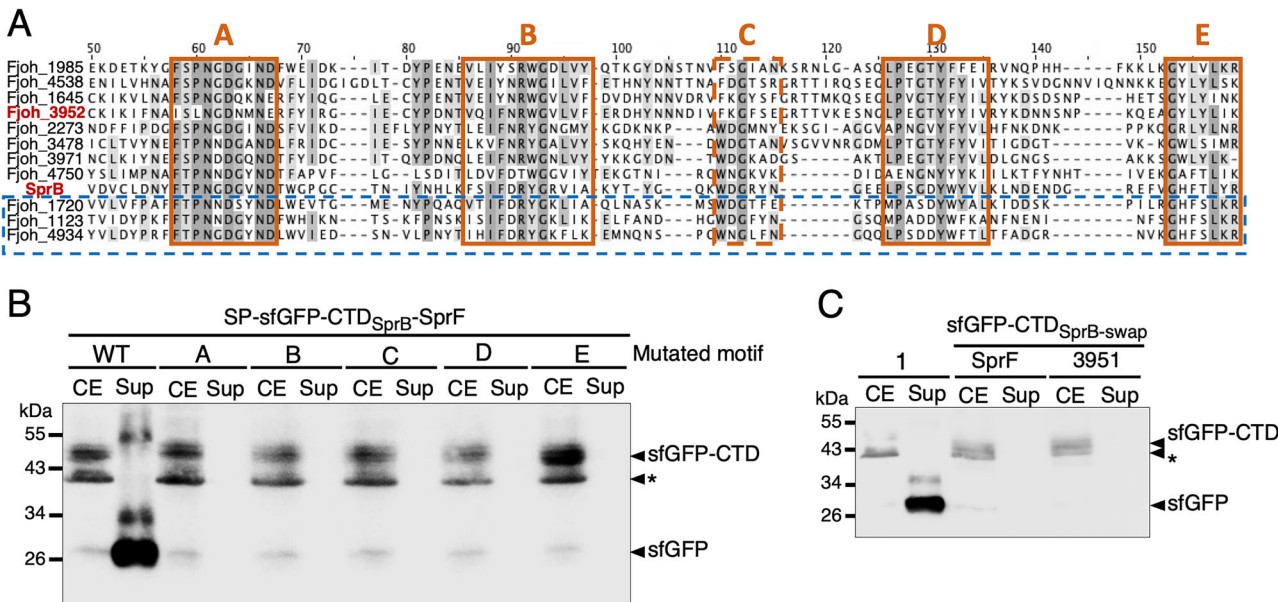

**Fig. 5 | Type B CTD conserved motifs are necessary for secretion but not sufficient for recognition by the cognate shuttle protein.** **A** Sequence alignment of the last 100 amino acids of Type B CTDs found in *F. johnsoniae* revealed five conserved motifs (A, B, C, D, and E). Alignments were performed using ClustalOmega (Madeira et al.[56]). The three orphan CTDs are boxed in blue dash lines. The CTDs of SprB and Fjoh_3952 are shown in bold red. **B** The secretion of sfGFP-CTD$_{SprB}$, in which each of the five motifs was independently mutated, was monitored when co-expressed with SprF. **C** The secretion of sfGFP-CTD$_{SprB-swap}$, in which all five motifs were simultaneously replaced by the corresponding motifs from Fjoh_3952, was monitored when co-expressed with either SprF or Fjoh_3951. Total cell extract (CE) and supernatant (Sup) were separated and analyzed by anti-GFP immunoblotting. The asterisk (*) indicates a non-specific band detected by the anti-GFP antibody batch used. Protein molecular mass (kDa) is shown on the left. Uncropped blots are shown in Supplementary Fig. 8.

To test whether the CTDs constitute the targeting signal and are responsible for the effector's localization and dynamics, we engineered chimeras between these three substrates. First, the native CTDs of AmyB and RemA were swapped with that of SprB. Both AmyB-HaloTag-CTD$_{SprB}$ and RemA-HaloTag-CTD$_{SprB}$ fusions were efficiently secreted and moved along the cell body, following a helical path similar to the SprB helical pattern (Fig. 6B, *middle and right panels*, Supplementary Movie 4 and 5[44]). These data demonstrate that the Type B CTD of SprB is necessary and sufficient to direct different substrates, an adhesin or a globular enzyme, to the gliding machinery. In a reciprocal experiment, the Type B CTD of SprB was swapped by the Type A CTD of RemA or of AmyB. SprB-HaloTag-CTD$_{RemA}$ was efficiently secreted and exposed to the cell surface, but remained immobile with, in some instance, disappearance of the signal (Fig. 6C, *right panel*, Supplementary Movie 6[44]), suggesting anchoring to the cell surface and possibly release into the medium (Fig. 6E). Figure 6D shows that most cells producing SprB-HaloTag-CTD$_{AmyB}$ displayed no fluorescence, and in the rare case of the detection of a fixed fluorescent signal, it quickly disappeared, suggesting secretion in the medium. Indeed, dot blot assays confirmed efficient secretion of SprB-HaloTag when fused to the CTD of AmyB (Fig. 6E). Taken together, these results show that chimera's localization and dynamics are dictated by the CTD, which is sufficient to target the substrate after exit from the translocon. We conclude that the Type A CTD supports either static anchoring to the cell surface or release into the medium, in agreement with the hypothesis that the PorV shuttle is recycled after targeting the substrate to its final destination. By contrast, the Type B CTD of SprB seems to address any substrate to the gliding machinery, raising the question of whether Type B CTDs are specifically dedicated to this function.

To answer this question, we sought to define the final destination of three randomly selected Type B CTD substrates, Fjoh_1123, Fjoh_3952, and Fjoh_4750. The HaloTag was inserted directly upstream of the substrate's CTD, at its native chromosomal locus. Fjoh_4750-HaloTag-CTD fusion displayed a behavior similar to that of the SprB-HaloTag-CTD fusion, moving along the cell body axis, following a helical path (Fig. 7, *left panel*,

Supplementary Movie 7[44]), indicating that the CTD of Fjoh_4750 targets the adhesin to the gliding machinery after translocation across the outer membrane. Unfortunately, no fluorescence signal was observed with Fjoh_1123-HaloTag-CTD, possibly due to a very low expression level (Supplementary Fig. 5). Since CTDs are sufficient to target substrates to their proper location, we engineered the SprB-HaloTag-CTD$_{Fjoh\_1123}$ fusion. This construct, expressed from the *sprB* operon promoter, served as a proxy to assess the location and dynamics of Fjoh_1123. SprB-HaloTag-CTD$_{Fjoh\_1123}$ exhibited two different behaviors. While most foci were static (Fig. 7, *middle left panel*), some of them (observed in 3% of the cells, $n = 525$) moved along the cell, following a helical track, such as SprB and Fjoh_4750 (Fig. 7, *middle right panel*, Supplementary Movie 8[44]), indicating that the CTD of Fjoh_1123 can target a subpopulation of adhesins to the gliding machinery. As opposed to the two previous fusions, Fjoh_3952-HaloTag-CTD and SprB-HaloTag-CTD$_{Fjoh\_3952}$ fusions only displayed static foci (Fig. 7, *right panel*, Supplementary Fig. 5, Supplementary Movie 9[44]), suggesting that the purpose of the Fjoh_3952 CTD is not to target the adhesin to the gliding machinery. In conclusion, our results demonstrate that Type B substrates are not all directed to the gliding machinery. Some Type B CTDs target their putative adhesin to static locations, possibly for cell-cell or cell-surface adhesion.

## Discussion

Here, we characterized a family of T9SS-dependent secretion signals called Type B CTDs and their associated shuttle proteins, necessary for the exit from the T9SS outer membrane translocon and targeting to the cell surface. Consistent with the work of Kulkarni and colleagues[21], our results show that Type B CTDs and PorP/SprF-like shuttle proteins form cognate pairs with strong specificity. At the molecular level, this specificity likely relies on five conserved motifs, which are necessary but not sufficient for proper recognition. Furthermore, our study revealed that three orphan Type B CTDs, Fjoh_1123, Fjoh_1720, and Fjoh_4934 are recognized by the same shuttle orphan protein, Fjoh_1677. This result also emphasizes that crosstalk may occur between Type B CTDs and PorP/SprF-like shuttle proteins, which we

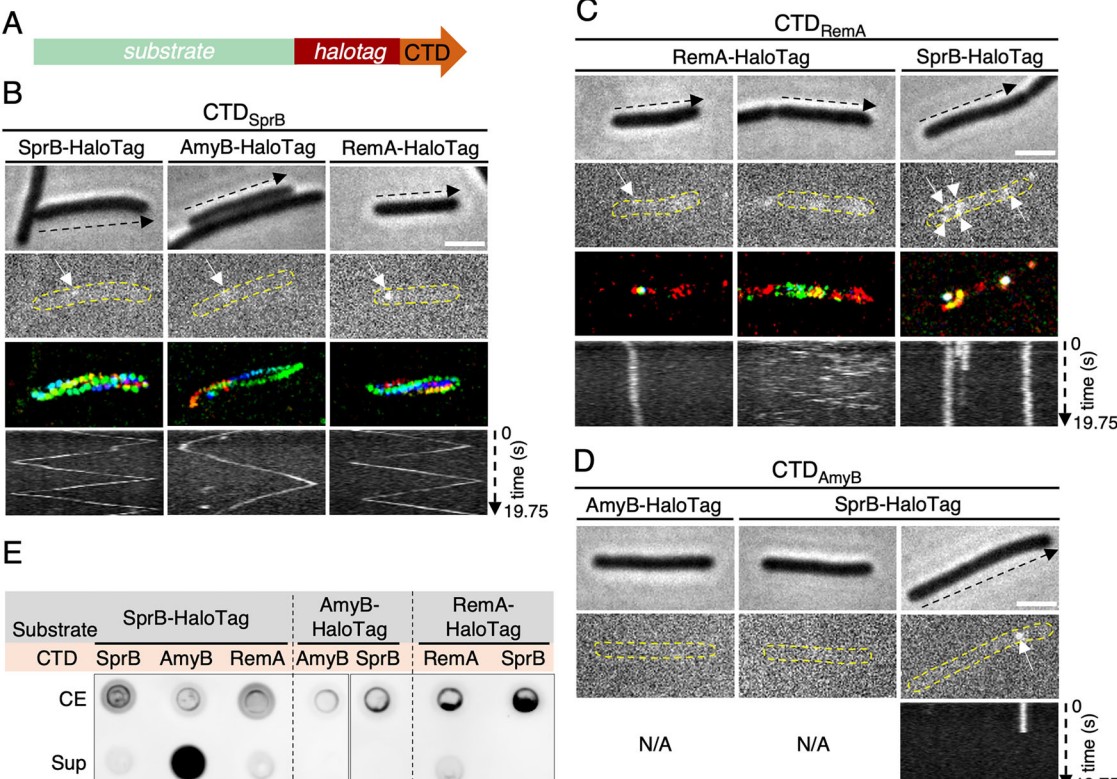

**Fig. 6 | CTDs determine the localization of their substrate after secretion.**
**A** Schematic representation showing that HaloTag is inserted directly upstream of
the CTD in the fusion proteins tested. To construct chimera fusions, CTDs were
swapped. **B–D** Localization and behavior of HaloTag fusions. For all strains, cells
were sandwiched between an agarose pad (2%) and a glass coverslip to significantly
limit cell movement to facilitate fluorescence signal acquisition and analysis.
Fluorescence was recorded at 250 ms intervals for several seconds. The phase con-
trast image (top panel), the first frame (second panel), and the kymograph of the
fluorescence signal (bottom panel) are shown. When no fluorescence was detected,
the kymograph was not generated and is marked N/A (Non-applicable) in the figure.
Scale bar, 2 μm. **B, C** The third panel shows the rainbow trace generated by the
motion of the fusions over time. **B** The movement of AmyB-HaloTag ($n = 75$) and
RemA-HaloTag ($n = 81$) fused to the CTD of SprB was monitored and compared to
the movement of SprB-HaloTag-CTD$_{SprB}$ ($n = 89$). **C** The localization and behavior

of the SprB-HaloTag-CTD$_{RemA}$ fusion ($n = 78$) was followed and compared to the
localization and dynamics of RemA-HaloTag-CTD$_{RemA}$ fusion ($n = 86$). A sub-
population of foci corresponding to RemA-HaloTag-CTD$_{RemA}$ fusions displays a
static behavior. Another subpopulation displays a highly dynamic behavior but does
not follow a helical trajectory. **D** No fluorescent signal was detected in the strain
producing AmyB-HaloTag-CTD$_{AmyB}$ fusion ($n = 76$). Few foci quickly disappearing
over time are detected in the strain producing the SprB-HaloTag-CTD$_{AmyB}$ fusion
($n = 90$). For each fusion, a representative cell is shown. **E** Dot blot analysis of
secretion assays with the fusions revealed that the CTD of AmyB allows efficient
release of SprB-HaloTag in the supernatant, that the CTD of RemA allows both
anchoring and release of SprB-HaloTag, and that the CTD of SprB allows both
AmyB-Halotag and RemA-Halotag Type A substrates to be anchored at the cell
surface. Uncropped blots are shown in Supplementary Fig. 8.

further demonstrated in the case of CTDs sharing high level of sequence
identity.

Type A CTDs generally share a common Ig-like structure, despite low
sequence identity (<40%)[39]. Previous studies proposed that the recognition
of Type A CTDs by T9SS components, notably by the PorV shuttle, depends
on conserved structural motifs rather than their sequence[11,39]. Indeed,
Lauber and colleagues suggested that the interactions between PorV and
Type A CTDs are predominantly hydrophobic interactions and that plas-
ticity allowed fine remodeling to accommodate slightly varying CTDs[13]. In
the case of Type B CTDs, despite similar or higher sequence identity
compared to Type A CTDs (Table 1) and conserved structural fold (Sup-
plementary Fig. 1), there is much higher specificity in the recognition
between CTDs and their cognate shuttle proteins. This suggests that con-
served motifs or residue side chains are involved in this specific recognition.
We showed that five conserved motifs (A to E) are essential but not sufficient
for this recognition and for specificity. These results provide a first basis for
understanding the recognition mechanism of Type B CTDs by their specific
shuttles. Further work is needed to map the residues/motifs involved in the
specific interactions.

Interestingly, we observed several cases of crosstalk. First, the three
orphan Type B substrates, Fjoh_1123, Fjoh_1720, and Fjoh_4934, use the

same shuttle protein, Fjoh_1667, even though their CTDs are not identical
(Fig. 8). Second, we showed that the CTD of Fjoh_3952 can be handled by its
cognate shuttle, Fjoh_3951, but also by Fjoh_1646, the shuttle pairing with
Fjoh_1645 (Fig. 8). Hence, Type B CTD/shuttle pairing is not strict. We may
hypothesize that these CTDs share common features recognized by the
shuttle(s) protein(s). Further studies will determine whether substrate
crosstalk also correlates with the biological function of these putative
adhesins.

We investigated if specific pairing between Type B substrates and
shuttle proteins may have a biological relevance. Do shuttle proteins
control the final destination and fate of the substrates? We showed that
the CTD of the gliding adhesin SprB is sufficient to promote the secretion
and transfer of Type A substrates, AmyB and RemA, to the gliding
machinery (Fig. 6B). Conversely, we demonstrated that Type A CTDs
support the secretion of the adhesin SprB but alter its final destination
and function by either releasing it into the medium or attaching it to the
cell surface, depending on the CTD (Fig. 6C, D). These findings indicate
that each CTD encodes instructions both for exiting the SprA translocon
and reaching its appropriate post-secretion destination. Interestingly, we
observed two distinct behaviors, likely linked to two different functions.
The CTDs of SprB, Fjoh_4750, and Fjoh_1123 direct the substrates to the

**Fig. 7 | Not all Type B substrates are directed to the gliding machinery.** Localization and behavior of HaloTag chimera fusions. For all strains, cells were sandwiched between an agarose pad (2%) and a glass coverslip to significantly limit cell movement to facilitate fluorescence signal acquisition and analysis. Fluorescence was recorded at 250 ms intervals for several seconds. The phase contrast image (top panel), the first frame (second panel), and the kymograph of the fluorescence signal (bottom panel) are shown. The behavior of Fjoh_4750-HaloTag-CTD ($n = 80$), SprB-HaloTag-CTD_Fjoh_1123 ($n = 525$), and Fjoh_3952-HaloTag-CTD ($n = 83$) fusions was monitored. The third panel shows the rainbow trace generated by the motion of these fusions over time. Fjoh_4750-HaloTag-CTD, and a subpopulation of SprB-HaloTag-CTD_1123 foci followed a helical trajectory similar to that of SprB. For each fusion, a representative cell is shown. Scale bar, 2 µm.

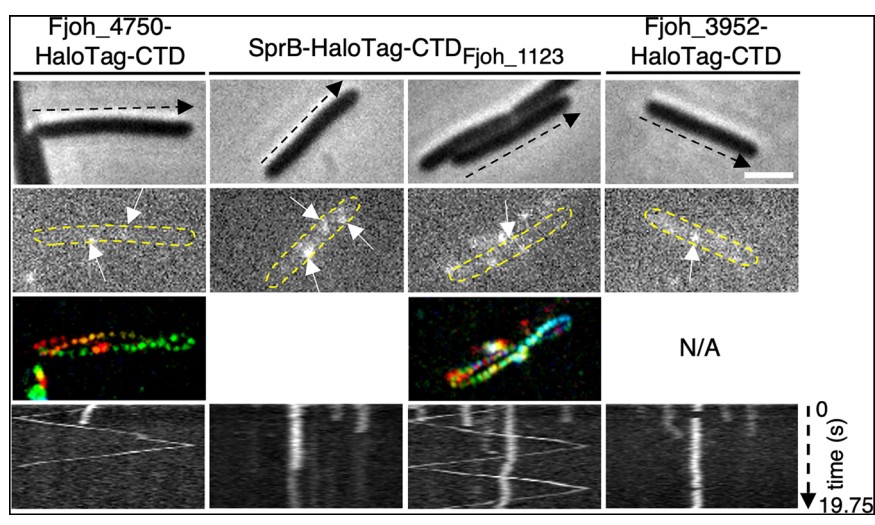

**Fig. 8 | A model of T9SS substrates targeted by the Type B CTD secretion signals.** The data presented in this study suggest that the Type B CTD secretion signal is both necessary and sufficient for secretion and for targeting substrates to their functional location. Type B substrates are extracted from the translocon by cognate shuttle proteins. Our results showed that the "orphans" substrates Fjoh_4934, Fjoh_1720, and Fjoh_1123 share the same shuttle Fjoh_1677. Furthermore, Fjoh_3952 can be handled by its cognate shuttle protein Fjoh_3951, but also by Fjoh_1646, indicating functional crosstalk between closely related CTDs. We also showed that a subset of Type B substrates (SprB, Fjoh_4750, Fjoh_4934, Fjoh_1720, and Fjoh_1123) is directed to the gliding machinery, while another subset of CTDs (Fjoh_3952) appears to be statically anchored to the cell wall by a PorE homolog (Fjoh_3950) and may be involved in adhesion function.

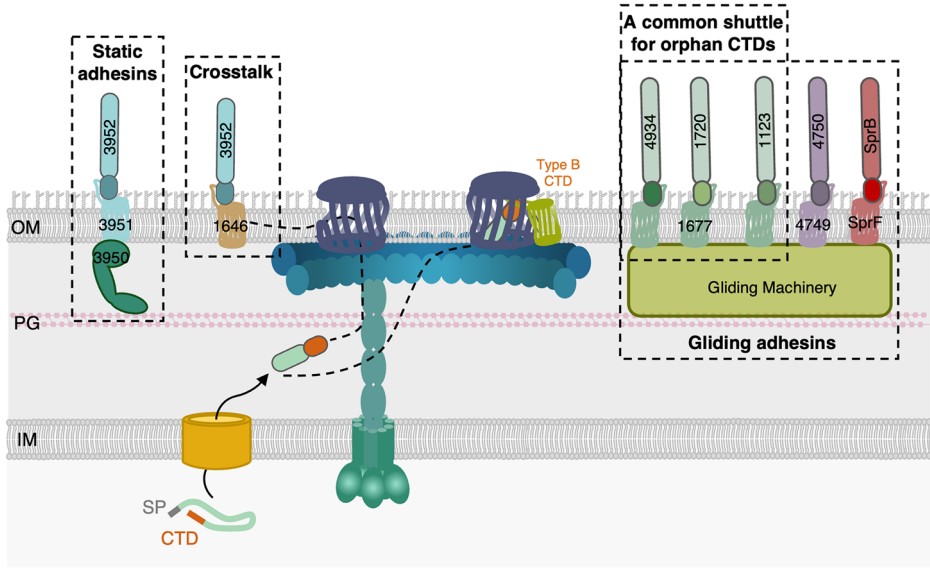

gliding machinery, allowing their helical displacement at the cell surface (Fig. 8). On the contrary, the CTD of Fjoh_3952 serves to anchor the substrate to the cell surface. This suggests that the CTD of Fjoh_3952, and possibly other similar Type B CTDs, are involved in attaching adhesins implicated in cell-cell or cell-surface adhesion rather than gliding. Both these functions are likely conserved in other bacteria using the T9SS for protein delivery. The human pathogen *P. gingivalis* possesses a single Type B substrate, PG1035, whose function remains unknown. Gorasia and colleagues reported that the Type B CTD of PG1035 interacts with PorP, which is also the unique PorP/SprF-like shuttle protein in *P. gingivalis*[41]. Based on the interaction of PorP with PorE, an outer membrane protein that binds to the peptidoglycan[45], they proposed that PG1035, PorP, and PorE form a trimeric complex anchored to the cell wall. Interestingly, the genome of *F. johnsoniae* harbors five operons, each of which contains genes encoding a PorE homolog, a Type B substrates, and a cognate SprF-like shuttle protein[21]. We can thus speculate that these proteins form a complex anchored to the cell wall in *F. johnsoniae*. Consistent with this idea, Fjoh_3950, encoded directly downstream of the gene encoding Fjoh_3951, is a PorE-like protein. Hence, Fjoh_3952 may interact with Fjoh_3951 and Fjoh_3950 in a static complex (Fig. 8), in agreement with our

fluorescence microscopy observations. Accordingly, dynamic substrates such as SprB, Fjoh_4750, and Fjoh_1123 are not encoded in operons with an associated PorE-like encoding gene.

In summary, Type B CTDs and their shuttle proteins form cognate pairs specified by conserved motifs, yet some degree of crosstalk exists. This pairing not only enables secretion but also determines the substrate's final localization. Our findings suggest that Type B CTDs encode distinct instructions either for movement via the gliding machinery or for static anchoring at the cell surface.

## Methods

### Bacterial strains, plasmids, and growth conditions
All strains with chromosomal constructs are listed in Supplementary Table 3. *Escherichia coli* DH5α and S17-1 λpir were used for cloning procedure and bacterial conjugation, respectively. *E. coli* cells were grown in Lysogeny Broth at 37 °C. *Flavobacterium johnsoniae* CJ1827, a streptomycin-resistant *rpsL2* derivative of ATCC 17061 (UW101), was used as model. *F. johnsoniae* cells were grown at 28 °C in Casitone Yeast Extract (CYE) medium[46]. For selection and maintenance, media were supplemented with antibiotics (erythromycin, 100 µg.mL$^{-1}$; streptomycin, 100 µg.mL$^{-1}$; tetracycline, 25 µg.mL$^{-1}$; kanamycin, 50 µg.mL$^{-1}$, and ampicillin, 100 µg.mL$^{-1}$).

## Genetic constructs

All plasmids and oligonucleotides primers used in this study are listed in Supplementary Tables 4 and 5, respectively. Enzymes for PCR and cloning were used as suggested by manufacturers. Chromosomal mutants and expression plasmids were generated as previously described[47] and are detailed in the supplementary material (Supplementary Methods, Supplementary Table 6).

## Precipitation of proteins in the supernatant

*F. johnsoniae* cells were grown in CYE at 28 °C with shaking to an absorbance at $\lambda = 600$ nm ($A_{600}$) of 0.8. For whole-cell samples, 1 mL of cells were pelleted by centrifugation at $10,000 \times g$ for 1 min. The supernatant was discarded and cells were resuspended in 100 µL of Laemmli loading buffer and boiled for 10 min. Proteins from the supernatant were precipitated as follows.

1.4 mL of culture were centrifuged at $20,000 \times g$ for 1 min to separate the supernatant from the cell pellet. 1.2 mL of supernatant was retrieved and subjected to a second centrifugation at $20,000 \times g$ for 1 min to discard contaminating cells. 1 mL of supernatant was transferred to a fresh tube, and proteins were precipitated by addition of trichloroacetic acid (TCA; 15% final concentration) for 30 min on ice. Precipitated proteins were recovered by centrifugation at $20,000 \times g$ at 4 °C for 20 min. Protein pellets were washed with 400 µL of ice-cold acetone, air dried for 5 min, resuspended in 30 µL of Laemmli loading buffer, and boiled for 10 min before blot analyses. 10 µL (equivalent to 333 µL of supernatant) was loaded for Western blot analysis.

## Western blot and Dot blot analyses

For Western blot analysis, the equivalent of 100 µL of culture at $A_{600} = 1$ (approximately 10 µL of cell extract) or 10 µL of supernatant fractions (prepared as described in the previous section) was loaded. Proteins were separated by SDS-PAGE and transferred onto a nitrocellulose membrane. For Dot Blot analysis, 5 µL of each sample was directly spotted onto a nitrocellulose membrane. Anti-GFP[48], anti-HaloTag (Promega, Monoclonal antibody), and NbALFA (NanoTag Biotechnologies) antibodies were used at 1:10,000, 1:1000, and 1:5000 dilutions, respectively. Primary antibodies were detected using horseradish peroxidase-conjugated goat anti-rabbit G (Sigma-Aldrich), horseradish peroxidase-conjugated goat anti-mouse G (Invitrogen), or horseradish peroxidase-conjugated goat anti-Llama IgG (Bethyl Laboratories) at a 1:10,000 dilution, followed by detection with the Western Lightning™ Ultra kit (Revvity).

## Fluorescence microscopy and image analysis

Five mL of cells were grown in CYE at 28 °C with shaking to an $A_{600}$ of 0.8. 500 µL of cells were pelleted and resuspended in 50 µL of CYE containing 200, 20, 2, or 0.2 nM of Janelia Fluor 549 HaloTag Ligand (Promega), depending on the strain. After incubation for 5 min, cells were washed once with CYE and 1 µL was spotted onto pads made of CYE supplemented with 2% of low-gelling temperature agarose (Sigma A9045). Phase contrast and fluorescence were recorded in oblique illumination mode (HiLo) on a Nikon Eclipse Ti2 microscope equipped with a 100 × NA 1.45 Ph3 objective, an Ilas2 system (Gataca Systems) and an Orca-Fusion digital camera (Hamamatsu). Images were analyzed using ImageJ (https://imagej.net/). Kymographs were generated using the KymoResliceWide plugin (https://imagej.net/plugins/kymoreslicewide).

## Cross-linking and co-immunoprecipitation

Co-immunoprecipitation assay was performed as previously described[41]. 200 mL of *F. johnsoniae* cells producing sfGFP-CTD fusions under the control of the *remA* promoter were grown in CYE at 28°C with shaking to an $A_{600}$ of 0.8. Cells were then washed twice with ice-cold PBS and concentrated 100-fold. Disuccinimidyl sulfoxide (DSSO) cross-linker was added to a final concentration of 1 mM, and the reaction was incubated 15 min at room temperature. The reaction was quenched by adding Tris-HCl pH 8.0 to a final concentration of 20 mM. Cells were washed twice with ice-cold PBS to remove excess of non-reacted cross-linker. Cells were then resuspended in 20 mM Tris-HCl, pH 8.0, 100 mM NaCl, 1% n-dodecyl-β-D-maltoside (DDM, Avanti), and protease inhibitor (cOmplete, Roche) and sonicated on ice (force 8; duty cycle 80%; 3 pulses of 20 s, Branson Ultrasonics Sonifier 450). After clarification by centrifugation at $16,000 \times g$ for 30 min at 4 °C, cell lysates were mixed with GFP-Trap agarose (Chromotek) and incubated 2 h on a wheel at 4°C. Beads were washed three times with Wash buffer (20 mM Tris-HCl pH 8.0, 100 mM NaCl, 0.1% DDM, and cOmplete protease inhibitor) before resuspension in 50 µL of Laemmli buffer and boiled for 10 min.

## Mass spectrometry analysis

Pull-down proteins were loaded on NuPAGE™ 4–12% Bis–tris Mini or Midi acrylamide gels according to the manufacturer's instructions (Invitrogen, Life Technologies). Running of samples was stopped as soon as proteins entered into the upper part of the gel and was stacked in a single band (80 V, 7 min). Protein-containing bands were stained with Thermo Scientific Imperial Blue, cut from the gel, and following reduction and iodoacetamide alkylation, digested with high sequencing grade trypsin (Promega). Extracted peptides were concentrated before mass spectrometry analysis under speed-vacuum. Samples were reconstituted with 0.1% trifluoroacetic acid in 2% acetonitrile and analyzed by liquid chromatography (LC)-tandem MS (MS/MS) using a Q Exactive Plus Hybrid Quadrupole-Orbitrap online with a nanoLC Ultimate 3000 chromatography system (Thermo Fisher Scientific). For each biological sample, 3 µL corresponding to 20% of digested sample were injected in triplicate on the system. After preconcentration and washing of the sample on a Acclaim PepMap 100 column (C18, 2 cm × 100 µm i.d. 100 Å pore size, 5 µm particle size), peptides were separated on a LC EASY-Spray column (C18, 50 cm × 75 µm i.d., 100 Å, 2 µm, 100 Å particle size) at a flow rate of 300 nL.min⁻¹ with a two-step linear gradient (2–22% acetonitrile/H₂0; 0.1% formic acid for 100 min and 22–32% acetonitrile/H₂0; 0.1% formic acid for 20 min). For peptides ionization in the EASY-Spray source, spray voltage was set at 1.9 kV and the capillary temperature at 250 °C. All samples were measured in a data-dependent acquisition mode. Each run was preceded by a blank MS run in order to monitor system background. The peptide masses were measured in a survey full scan (scan range 375–1500 $m/z$, with 70 K FWHM resolution at $m/z = 400$, target AGC value of $3.00 \times 10^6$ and maximum injection time of 100 ms). Following the high-resolution full scan in the Orbitrap, the 10 most intense data-dependent precursor ions were successively fragmented in HCD cell and measured in Orbitrap (normalized collision energy of 25%, activation time of 10 ms, target AGC value of $1.00 \times 10^5$, intensity threshold $1.00 \times 10^4$ maximum injection time 100 ms, isolation window 2 $m/z$, 17.5 K FWHM resolution, scan range 200 to 2000 $m/z$). Dynamic exclusion was implemented with a repeat count of 1 and exclusion duration of 20 s.

## Protein identification and quantification

Relative intensity-based label-free quantification (LFQ) was processed using the MaxLFQ algorithm[49] from the freely available MaxQuant computational proteomics platform, version 1.6.3.4[50]. Analysis was done on three biological replicates, each injected three times on mass spectrometers. The acquired raw LC Orbitrap MS data were first processed using the integrated Andromeda search engine[51]. Spectra were searched against the *F. johnsoniae* database extracted from UniProt (UP000006694, date 2022 May 4th; 5016 entries)[52]. The false discovery rate (FDR) at the peptide and protein levels were set to 1% and determined by searching a reverse database. For protein grouping, all proteins that could not be distinguished based on their identified peptides were assembled into a single entry according to the MaxQuant rules. The statistical analysis was done with Perseus program (version 1.6.15) from the MaxQuant environment (www.maxquant.org)[53]. Quantifiable proteins were defined as those detected in above 70% of samples in one condition or more. Protein LFQ normalized intensities were base 2

logarithmized to obtain a normal distribution. Missing values were replaced using data imputation by randomly selecting from a normal distribution centered on the lower edge of the intensity values that simulates signals of low-abundant proteins using default parameters (a downshift of 1.8 standard deviation and a width of 0.3 of the original distribution). To determine whether a given detected protein was specifically differential between two conditions, a two-sample $t$-test was done using permutation-based FDR-controlled at 5 and employing 250 permutations. The $p$-value was adjusted using a scaling factor s0 with a value of 1.

The mass spectrometry proteomics data have been deposited to the ProteomeXchange Consortium via the PRIDE partner repository with the dataset identifier PXD065655[54,55].

## Statistics and reproducibility
All microscopy experiments were performed using three independent biological replicates. For each replicate, representative cells were selected and analyzed, resulting in the following total numbers of cells: AmyB-HaloTag (n = 75), RemA-HaloTag (n = 81), SprB-HaloTag-CTDSprB (n = 89), SprB-HaloTag-CTDRemA fusion (n = 78), RemA-HaloTag-CTDRemA fusion (n = 86), AmyB-HaloTag-CTDAmyB fusion (n = 76), SprB-HaloTag-CTDAmyB fusion (n = 90), Fjoh_4750-HaloTag-CTD (n = 80), SprB-HaloTag-CTDFjoh_1123 (n = 525), and Fjoh_3952-HaloTag-CTD (n = 83). Co-immunoprecipitation assays were performed in three independent biological replicates, and each sample was analyzed by mass spectrometry in technical triplicates.

## Reporting summary
Further information on research design is available in the Nature Portfolio Reporting Summary linked to this article.

## Data availability
The movies and Supplementary Data have been deposited on Figshare: https://doi.org/10.6084/m9.figshare.30047764.v1. The mass spectrometry proteomics data have been deposited to the ProteomeXchange Consortium via the PRIDE partner repository with the dataset identifier PXD065655. Numerical data used to generate the graphs in Fig. 2 are provided in Supplementary Data.

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

## Acknowledgements

The authors would like to thank the members of the Cascales research group for support and insightful discussions, in particular Yaëlle Aouizerate. The authors thank Melissa Guilbert, Fabien Gomez, and Audrey Gozzi for technical assistance, and Daisy Dratey for encouragement. This work was supported by Aix-Marseille Université (amU), the Centre National de la Recherche Scientifique (CNRS), and grants from the Agence Nationale de la Recherche (ANR-20-CE11-0011 and ANR-23-CE11-0035) and from the Excellence Initiative of Aix-Marseille University (A*MIDEX, A-M-AAP-ID-17-33-170301-07.22). M.P. is supported by a doctoral fellowship from the French Ministère de l'Enseignement Supérieur, de la Recherche et de l'Innovation and by an end-of-thesis fellowship from the Fondation pour la Recherche Médicale (FDT202404018242). Marseille Proteomics (marseille-proteomique.univ-amu.fr/) was supported by Institut Paoli-Calmettes (IPC), Région FEDER, and INSERM funding, Canceropôle PACA, IBISA (Infrastructures Biologie Santé et Agronomie), and Plateforme Technologique Aix-Marseille University.

## Author contributions

Conceptualization of project: M.P., E.C. and T.D.; methodology: M.P., C.C.H., S.A., E.C. and T.D.; investigation: M.P., S.A. and T.D.; funding acquisition: E.C. and T.D.; Project administration: T.D.; supervision: E.C. and T.D.; writing—original draft: M.P. and T.D.; writing—review and editing: M.P., E.C. and T.D. All authors commented on the final version of the manuscript.

## Competing interests

The authors declare no competing interests.
