## [Transparent Peer Review file · Communications Biology]

Specialized shuttle proteins recognize Type IX secretion signals and target effectors to their final destinations in *Flavobacterium johnsoniae*

Corresponding Author: Dr Thierry Doan

Version 0:

Reviewer comments:

Reviewer #1

(Remarks to the Author)

I had the pleasure of reviewing the manuscript. Here is my review of it.

Brief summary of the manuscript:

This work identifies a crucial partner protein necessary for the secretion of canonically orphan members within a specific subtype of proteins secreted by T9SS. It also highlights conserved motifs in the recognition domain of this cargo protein subtype and demonstrates their significance. Finally, the study reveals that secretion via this domain subtype does not follow an expected rule relating to the cargo's destination, challenging what could have been an intuitive assumption to make.

Overall Impression:

This is an interesting and well-structured study that systematically addresses important questions in an intuitive, curiosity-driven manner. Each claim is supported by strong experimental evidence. The work advances our understanding of how T9SS- the protein sorting mechanism found in a herd of important and interesting bacteria functions.

Specific comments:

Line 20: The sentence omits recognition of ChiA as a type of CTD that is neither Type A nor Type B; it is recommended to rephrase, even if ChiA CTD is not explicitly mentioned.

Line 64: "...where they are folded and recruited to the T9SS." – To the best of my knowledge, there is no experimental evidence demonstrating whether all or some T9SS substrates are secreted in a folded or unfolded state. The authors must add a citation to support this statement or rephrase it.

Line 65: "...conserved 100-amino-acid C-terminal domain (CTD)..." – CTD-B is longer. Kulkarni et al. (2019) experimentally validated that the required length of C terminal amino acids for Type B CTD mediated secretion is between ~150-220 amino acids. While the sequence similarities span a shorter stretch of ~100 amino acids, the 'functional CTD' extends beyond this. I am not in immediate recollection of whether predicted folding structures of the extended C-terminal regions follow the conservation as the ~100aa-long regions do. However, given it is functional at that length, and assuming that it does conserve the folding across Type-B CTD cargo proteins, it may be more accurate to refer to the broader region as the CTD-B. But anyhow, I suggest, authors indicate if not a broader and more inclusive range, they could go with something more accurate than a categorical "100" amino acids.

Findings of Figure 1: The discoveries presented in the first figure are not novel to this paper. As acknowledged by the authors, these findings have been previously reported, and even the hypothesis stated in Line 115—"we hypothesized that the amount of endogenous PorP/SprF proteins produced is insufficient to ensure the efficient secretion of both the native Type B substrate and the corresponding highly produced sfGFP CTD fusion"—has been previously proposed as an explanation for the same observation (Kulkarni et al., 2019). While this foundational information is important for contextualizing the work, the authors should consider whether Figure 1 and the first section of the Results would be more appropriate in the Supplementary Data.

Line 128: The term "linked" may not be the most accurate choice here. "Adjacent" might be more suitable, as "linked" could

imply a functional or transcriptional connection, such as co-transcription as a single ORF, for which no current evidence exists.

Line 153: "These results suggest that Fjoh_1677 is the shuttle protein." – This may be premature. The experiment establishes an interaction, and in this context, suggests that Fjoh_1677 may be the shuttle protein, but additional evidence is needed for a definitive conclusion.

Line 171: "These alignments also showed that, while the vast majority of sequence identity between Type B CTDs is 20-30%, other Type B CTDs share >50% sequence identity (Table 1), suggesting that cross-talks might be possible between these CTD/shuttle pairs." – This sentence is unclear and should be rephrased to explicitly clarify which groups are being compared.

Line 179: "...-specific shuttles" – The listed proteins are shuttles, not cargos. This appears to be an error. (for eg: "... Fjoh_3952, or with the Fjoh_1645, Fjoh_3971 and SprB - specific shuttles" would be correct)

Line 185: In addition to sequence similarity, the alignment of AlphaFold structures deserves mention here. A brief note on structural comparisons would add valuable insight.

Line 192: While limiting the comparison to the last 100 amino acids of the CTD may be practical, have the authors considered that the functional Type B CTD extends to ~150-220 amino acids? Given this, it may be worth conducting multiple sequence alignments on the entire CTD region to identify additional conserved motifs.

Line 197: "B3942" – This appears to be a typo.

Line 229: "86% of the cells" – A typo?

Line 230: How exactly does the motion of RemA-Halo differ from RemA-CTD? The figure legend also does not describe the motion. A brief statement clarifying whether the motion is helical or follows another pattern would be useful. Additionally, providing a link to a supplementary video would enhance visualization for the reader. Since RemA was found to move helically on the cell surface similarly to SprB (Shrivastava et al., 2012), and the findings here differ, this discrepancy should be discussed.

Figures 6 and 7: The number of cells observed should be stated in the legend, even though it is mentioned in the main text.

Line 253: "By contrast, the Type B CTD of SprB addresses any substrate to the gliding machinery." – A citation should be added here.

Line 357: Precipitation of extracellular proteins – What volume of culture was taken? This is crucial information for reproducibility and is missing in other experiments as well, such as in the Cross-linking and co-immunoprecipitation.

Line 368: Western Blot and Dot Blot Analyses – Missing details: How much protein was loaded per well? What was the starting volume of culture for protein harvesting? These details should be included for reproducibility.

Line 396: Sonication parameters should be specified, as they are crucial determinants of the experiment's outcome. Additionally, DDM is mentioned without its full form—this should be corrected.

Figure 1:

The schematic should specify which protein's CTD is being tested in this experiment or at least indicate that it is a CTD-B. It would also be helpful to mention the presence of a Sec signal peptide at the N-terminal end, especially since it is referenced in panel B.

The legend states:

"SprB and Fjoh_3952, with and without coexpression of SprF (SprB-SprF) and Fjoh_3951 (3952-3951) respectively," but the textual annotation above the gel does not clearly indicate where the "without SprF" data is shown, if shown at all. The formatting of this panel may need adjustment to make it clearer which lanes correspond to CTD-B and whether a cognate shuttle is co-expressed. Ensuring this will make the figure more technically accurate and accessible, especially for junior researchers and those from outside the field.

Additionally, the legend should explicitly mention that "SP" refers to "signal peptide." Also, where sfGFP_{peri} is mentioned, it is technically incorrect since this construct does not contain any CTD.

Figure 2:

The letters following "CTD" should be subscripted for clarity and consistency.

- The base of the logarithm used should be specified in the figure, as the Methods section mentions a base of 2—this is worth clarifying to avoid confusion.

- The legend states:

"When the protein was absent from the sfGFP_{peri} strain, an arbitrary value was assigned to calculate a ratio, and the bar has been colored grey."

The Methods section should further elaborate on how this computation was performed. Specifically:

- How was this "arbitrary value" chosen?
- Does it vary across different CTDs, such as in subpanels E and F?
- Does the choice of this arbitrary value affect the ranking of the candidate protein (Fjoh_1677) on the x-axis?
- Is using an arbitrary value a standard approach in this type of analysis, or could a fixed value, such as "1" or its equivalent in log-transformed form, be used instead for normalization?

Clarifying these points would strengthen the methodological transparency of the analysis.

Figure 3:

The authors should consider consolidating two panels into one, grouping data as CTD-without-shuttle – CTD-with-shuttle pairs (e.g., 1123 | 1123+1677, 1720 | 1720+1677 ...). This would significantly improve figure readability.

If the current layout is retained, Panel A should also include a schematic similar to Panel B, effectively recapitulating part of Figure 1 to provide better context for the reader.

Figure 4:

The "null set" symbol should be replaced with the stain annotation and the figure legend should describe the subsets.

Reviewer #2

(Remarks to the Author)

Manuscript ID: COMMSBIO-25-0230-T

Title: Specialized shuttle proteins recognize and address type IX secretion effectors

Main comment:

The authors have performed an interesting study regarding the Type B CTD/ shuttle protein interaction and secretion in Type IX secretion system. Multidisciplinary methodologies are combined to tackle fundamental questions and draw interesting conclusions/hypothesis, expanding our knowledge in the field of type IX.

However, the quality of the research is not well translated in the manuscript and several key points need significant improvement regarding the clarity, organization and data presentation within the manuscript. Description of the results and figures are too elusive; the manuscript is written in a rush and not well structured leading to a not concise presentation of the main arguments.

After careful consideration, I am unable to recommend this paper for immediate publication, unless the comments below are addressed and the new manuscript be reviewed again.

Major comments:

1: the manuscript is poorly written, and consequently lacks of consistency. As a reader, I feel like different sections have been written by different people and simply merged together (see lines 125-156, written in a completely different style than the rest of the manuscript). A lot of grammar, typos, spelling mistakes are present (some are listed below); and the phrasing used by the authors is misleading since the interpretation is not clearly connected to the presented data. Additionally, experimental procedures are poorly described and table legends are missing. As a reviewer, I suggest that the authors seek the assistance of a native English speaker to improve the overall quality of the manuscript.

Some examples of inconsistency across the manuscript:

a) Table S1 and S2 legends are missing, and although are called S1 and S2 in the main text, in the supplementary files they are called "strains", "Plasmids".

b) The quality of the figure seems to be below the requirement of the journal (the resolution and formatting need revision). This might be an artefact of the generated files of course.

c) Line 97, 366 and 368: should be "analysis" instead of "analyses"

d) line 405: should be "7 min" instead of "7 mn"

Many more examples could be provided; the manuscript need to be properly edited.

2: The names of the secretory proteins and protein derivatives used in the study are long and complicated, confusing the reader and breaking the flow of the paper. The authors could include a schematic representation of the system including proteins and engineered derivatives. It could be either in the main figures, or added in the supplementary materials. A concluding cartoon/ model showing chaperones binding (or not) to secretory proteins would also be an additional value for the paper and very helpful for a non-expert reader to comprehend all the results described in the present study.

3: Lines 115-123, 160-168, Fig. 1 and 3: The authors discuss about limiting interactions between chaperones and secretory proteins due to limited amount of proteins produced intracellularly. However, there are no data demonstrating this hypothesis. Since the authors are growing bacteria and collect samples for their secretion assays, they should perform western blots analysis to quantify the amount of those proteins produced inside the bacteria. The authors might then demonstrate that the protein abundance is indeed responsible of the limited interactions between chaperones and secretory proteins, resulting in a defective secretion phenotype. Since this is one of the main statement of the paper controls experiments of gene expression levels are essential to convince the reader.

#4: The number of biological/ technical repeats for each experiment needs to be mentioned. Moreover, the statistical significance of the comparison of the MS data should be calculated and included in Fig. 2.

#5: In Fig. 2C, SprB fusion protein is missing, so I assume that the protein was not detected by MS. Indeed, protein identification using LC-MS, especially after in-gel digestion, could lead to no protein detection for various reasons (poor peptide fractionation, abundance, effect of ionization, or not stably produced protein). The authors need to consolidate their hypothesis, rule out experimental/technical limitations (the production of proteins needs to be demonstrated by gel analysis) before interpreting their data and mention it in the main text.

6: "Material and methods" section needs extensive revisions as critical technical details are lacking. As an example, the authors are complementing bacteria in trans with plasmids carrying genes for secretion and co-purification assays. However, there are no details on the gene expression conditions (inducer concentration, time, temperature ...). Again, I am wondering how the authors secure that protein over-production (higher than chromosomal levels I suppose) does not lead to un-physiologically relevant interactions: a chaperone might interact with a secretory protein only because the production level is so high that unspecific interactions are showing, or because they are co-produced. This should be discussed in both the results and discussion sections.

7#: The authors defined 5 conserved motifs after sequence alignments between the Type B CTDs. However, it is not clear why a protein would need all 5 motifs for shuttle recognition and secretion, if swapping just one is enough to abolish one or both functions. One should swap all or more motifs to shed more light to the interaction mechanism and provide insight on why those motifs are / are not necessary for efficient secretion. The authors should do those experiments to strengthen their current results and conclusions and provide additional value to the present manuscript.

Moreover, the reason why they focused only on motif B and E is not very clear. The authors mentioned “notable differences between these motifs” (line 193), however, they should elaborate on this statement more and justify their choice.

Minor comments:

1: Figures resolution and formatting is poor in the printed version. Authors should check if the figures meet the journal requirements

#2: In the last paragraph of the results (lines 202-274), the authors mentioned “CTD dictates protein “dynamics”. However, from the data shown (Fig. 6 and 7), one could suggest that CTD dictates protein localization and function. Protein dynamics as a term refers to protein domain/chain movements, structural alterations and conformational changes. This study does not contain data demonstrating changes in dynamics (i.e structural analysis, MD simulation). Therefore, the authors should be carefully and justify the use of “dynamics” or replace it with the word “function” or activity that seem more proper according to their data.

#3: In Text S1, the cloning description is very complicated. A table listing all the suicide plasmids generated in this study (mentioning the name of the plasmid, the PCR product/insert generated, the template used and the primers used) would consolidate the procedure and improve the presented manuscript.

4: line 11: keywords are missing.

#5: line 91-96: A reference is missing from this statement.

#6: line 211: consider changing “collect the substrate..” with target or escort or transfer

Reviewer #3

(Remarks to the Author)

In this manuscript, Paillat et al. have studied how substrates of the Type IX Secretion System are recognised by shuttle proteins following secretion, which facilitates correct localisation of these substrates. C-terminal domains (CTD) on these substrates are recognised by these shuttle proteins and while Type A CTD are well-characterised, less is known about how Type B CTD are involved in substrate localisation. It has previously been shown that Type B CTD are encoded upstream of their cognate shuttle protein, for which they are specific.

Using co-immunoprecipitation mass spectrometry, a single shuttle protein responsible for the localisation of three orphan Type B CTD proteins was identified. These had higher homology in the CTD compared to other Type B CTD substrates, which suggested a mechanism by which these three substrates could be recognised by the same shuttle protein. Sequence conservation analysis identified five motifs that were conserved in all Type B CTD found in this strain, and mutagenesis of some of these demonstrated they were essential for secretion. Finally, it was shown that the CTD alone determined the extracellular localisation of these substrates, being targeted to the gliding motility apparatus, cell surface anchored or released from the cell.

Altogether this work presents some interesting findings in regard to T9SS substrate targeting, however I have a few concerns which are detailed below.

Major comments

Figure 1: It is unclear to me why Fig.1 has been included in the results. From what I can tell, the first part of panel B has been published in Kulkarni et al., 2017 and the second part has been published in Kulkarni et al., 2019 and so, should be included in the introduction. If a novel finding is to be presented from this experiment, it should be the focus of the discussion in text. Currently, lines 113-115, discuss the reliance of Type B CTD for downstream encoded SprF-like proteins, which cannot be concluded from Fig. 1 as a negative control is absent. In addition, it is not clear what panel A is meant to convey to the reader, given that it has little relation to the proteins presented in panel B.

Line 186/299; Fig. 5: While the motif analysis in Fig. 5A is compelling, the subsequent experiments do not really address the importance of these motifs. Can the authors address why motifs A, C and D were not tested? In addition, it is not clear which sequence in the alignment represents SprB, or if it is present at all. It is therefore not clear what residues have actually been mutated by switching the B and E motifs. In addition, if the reason that only B and E were tested is because A, C, and D were identical between SprB and Fjoh_3952, then this would suggest that regions outside the conserved motifs are also important for shuttle specificity. Further clarity and evidence is required to support the importance of these Type B CTD motifs in recognition by shuttle proteins. Would it be possible to use AlphaFold to model the CTD with the cognate shuttle protein to give insight into what residues may be facilitating recognition of the CTD? This could then be used to either support why B and E are important in the process and give insight into what other residues in the CTD are involved.

Minor comments:

Figure 2: In most samples tested, the expressed fusion protein is detected as one of the most abundant species, however in Fig. 2C, SprB does not appear to be detected. Is it known why SprB cannot be detected?

Figure 3: The presence of the gene diagram is beneficial to the reader for clarity, however, should be included in both panel A and B, not B alone. This is also true for Fig. 1 where it is only beneficial for the reader if it provides a visual reference for what is present in the figures.

Figure 6-7: There are several panels missing from both Figures 6D and 7. Is the data not available or is there a reason it has

been excluded?

Figure 5: As mentioned above, if SprB is present in the alignment in panel A, can it be labelled so it is easier for the reader to identify. Also, in the legend for 5B and C, it implies the BE double mutation was also tested for SprF, which doesn't seem to be true.

Spelling and grammar:

Line 156 – “Cross-talks” should be cross-talk.

Line 181 – “Fjoh_3971 and SprF” should be ‘or’.

Line 221/257 – “upstream the...” should be ‘upstream of the...’

Line 253 – I don't think addresses is the correct word here. Perhaps ‘directs’ or ‘targets’

Reviewer #4

(Remarks to the Author)

The authors primarily investigated the substrate recognition and secretion mechanisms of the Type IX Secretion System (T9SS) in bacteria, particularly members of the Bacteroidetes phylum, with a focus on the function of Type B C-terminal domains (CTDs) and their dedicated shuttle proteins. In *Flavobacterium johnsoniae*, the authors discovered that among the 12 Type B substrates, three "orphan" substrates (Fjoh_1123, Fjoh_1720, and Fjoh_4934) were not genetically linked to known PorP/SprF-like genes. Through co-immunoprecipitation and mass spectrometry analysis, it was found that these three substrates share the same orphan shuttle protein, Fjoh_1677, suggesting that a single shuttle protein can recognize multiple substrates. By performing motif-swapping experiments, the authors identified that two conserved motifs (B and E) are necessary for the recognition of substrates by their specific shuttle proteins. Additionally, it was found that the CTD is sufficient to guide heterologous proteins (e.g., sfGFP) through T9SS secretion and determines the final localization of the substrates (e.g., dynamic adhesion to the gliding motility machinery or static anchoring to the cell surface). This study reveals the specific interaction mechanism between Type B CTDs and their dedicated shuttle proteins through conserved motifs, expanding our understanding of the diversity of bacterial secretion systems. The findings demonstrate that CTDs not only serve as secretion signals but also encode localization information, potentially regulating the functional division of substrates (e.g., motility, adhesion, or enzyme secretion) through different shuttle proteins. This provides a theoretical foundation for the engineering of bacterial secretion systems, such as the targeted delivery of proteins.

Major revision needs to be made:

1. In this study, co-immunoprecipitation (co-IP) and mass spectrometry (MS) were used to identify the shuttle protein responsible for the secretion of T9SS Type B CTD effector protein. But is there an independent experimental method (such as protein-protein interaction verification experiment, such as two-hybrid or surface plasmon resonance) to further confirm the specificity of these interactions?

Minor revisions need to be made:

1. Figure 1-3: The molecular weight markers in the Western blot diagram are not marked with specific values (only the kDa range is displayed), and some bands are blurred (as shown in the CE and Sup bands of Fjoh_4934 in Figure 3A). Need to provide higher resolution images and clear marks.
2. The format of some references is not uniform, so it needs to be uniform.
3. Mass spectrometry data was submitted to ProteomeXchange, but the login number was not provided (PXDxxxx is a placeholder). The actual identifier needs to be supplemented.
4. In line 32 change plays a crucial role for "to" "plays a crucial role in".
5. Type B CTDs can be divided into dynamic and static types. What is the biological significance of this functional differentiation? This could be discussed in discussion.
6. It is emphasized that Type B CTD is bifunctional (secretion+targeting), and more experimental data should be provided to support this view.
7. How to prove that CTD domain not only participates in secretion, but also determines the final location of protein?
8. It is suggested that the specific sequence of CTD determines the protein targeting, but can this be directly proved by deleting or replacing these sequences (such as motifs B and E)?
9. Will the level expression of Fjoh_1123-HaloTag-CTD have a certain impact?

Version 1:

Reviewer comments:

Reviewer #1

(Remarks to the Author)

Most of my comments are addressed satisfactorily except for the following.

1. Minor thing but, “Almost all CTDs are classified into Type A and Type B families.” does not sound too good either. Why

not rather say something along the lines of Type A and type B represent the major classes of

2. Minor point :: Line 110-111 :: "Endogenous locus" repeated

3. Fig6C- Is the pattern of movement in the mobile subset of remA consistent? If i squint my eyes to try to focus on the signal and avoid noise, I can see parts that appear helical in nature. This will be interesting and informing to visualize as a rainbow trace as in panel B. Authors should add this.

Reviewer #2

(Remarks to the Author)

Manuscript ID: COMMSBIO-25-0230-A

Title: Specialized shuttle proteins recognize Type IX secretion signals and target effectors to their final destinations.

In the present manuscript, the authors have performed an interesting study regarding the Type B CTD/ shuttle protein interaction and secretion in Type IX secretion system. They have combine multidisciplinary and up-to-date methodologies to address fundamental questions and draw interesting conclusions, expanding our knowledge in the field of type IX secretion system.

After extensive revision and substantial re-writing, the manuscript now demonstrates a clear and logical structure. The experimental procedures are well described, the results are robust and conclusive, and the figures and tables are appropriately formatted with comprehensive legends that enhance clarity.

In light of these improvements, I recommend that this manuscript be accepted for publication.

Reviewer #3

(Remarks to the Author)

The authors have addressed all of my comments and overall, have improved the quality of the manuscript during the review process.

One very minor comment I have is that the colours between proteins in the AlphaFold multimer models in Fig. S3A and S5D are not contrasting enough to clearly differentiate between the two proteins. Increasing the contrast would make the two proteins much easier to differentiate.

However, overall I have no further concerns and am happy to recommend the publication of this manuscript.

Reviewer #4

(Remarks to the Author)

Itemized responses to Reviewer's comments

COMMSBIO-25-0230-T

"Specialized shuttle proteins recognize and target type IX secretion effectors"
Paillat et al.

We sincerely thank all the reviewers for their valuable feedback. Their insightful comments helped us to significantly improve the manuscript. Point-to-point responses to their comments are listed below in blue.

In the tracked changes version provided for the main text, all changes are highlighted in yellow, together with line numbering.

Reviewers' comments:

Reviewer #1 (Remarks to the Author):

I had the pleasure of reviewing the manuscript. Here is my review of it.

Brief summary of the manuscript:

This work identifies a crucial partner protein necessary for the secretion of canonically orphan members within a specific subtype of proteins secreted by T9SS. It also highlights conserved motifs in the recognition domain of this cargo protein subtype and demonstrates their significance. Finally, the study reveals that secretion via this domain subtype does not follow an expected rule relating to the cargo's destination, challenging what could have been an intuitive assumption to make.

Overall Impression:

This is an interesting and well-structured study that systematically addresses important questions in an intuitive, curiosity-driven manner. Each claim is supported by strong experimental evidence. The work advances our understanding of how T9SS- the protein sorting mechanism found in a herd of important and interesting bacteria functions.

We thank the reviewer for the very positive appraisal of our work.

Specific comments:

Line 20: The sentence omits recognition of ChiA as a type of CTD that is neither Type A nor Type B; it is recommended to rephrase, even if ChiA CTD is not explicitly mentioned.

We modified the text (line 19 now) to take into account the idea that CTDs like the one of ChiA are neither Type A nor Type B, while keeping it simple for clarity:

“Almost all CTDs are classified into Type A and Type B families”

Line 64: “...where they are folded and recruited to the T9SS.” – To the best of my knowledge, there is no experimental evidence demonstrating whether all or some T9SS substrates are secreted in a folded or unfolded state. The authors must add a citation to support this statement or rephrase it.

It is indeed not clear whether substrates fold entirely or only partially in the periplasm. We modified the sentence (now line 56):

“All T9SS effectors possess an N-terminal Sec-dependent signal peptide, which supports their export to the periplasm, where they are recruited by the T9SS”

Line 65: “...conserved 100-amino-acid C-terminal domain (CTD)...” – CTD-B is longer. Kulkarni et al. (2019) experimentally validated that the required length of C terminal amino acids for Type B CTD mediated secretion is between ~150-220 amino acids. While the sequence similarities span a shorter stretch of ~100 amino acids, the ‘functional CTD’ extends beyond this. I am not in immediate recollection of whether predicted folding structures of the extended C-terminal regions follow the conservation as the ~100aa-long regions do. However, given it is functional at that length, and assuming that it does conserve the folding across Type-B CTD cargo proteins, it may be more accurate to refer to the broader region as the CTD-B. But anyhow, I suggest, authors indicate if not a broader and more inclusive range, they could go with something more accurate than a categorical “100” amino acids.

We modified the sentence to take into account this relevant comment (line 58 now):

“T9SS substrates recognition and sorting is mediated by a conserved C-terminal domain (CTD) that serves as a secretion signal (Veith et al., 2013), ranging from approximately 100 amino acids for Type A CTDs to 150-235 amino acids for Type B CTDs.”

Findings of Figure 1: The discoveries presented in the first figure are not novel to this paper. As acknowledged by the authors, these findings have been previously reported, and even the hypothesis stated in Line 115—“we hypothesized that the amount of endogenous PorP/SprF proteins produced is insufficient to ensure the efficient secretion of both the native Type B substrate and the corresponding highly produced sfGFP CTD fusion”—has been previously proposed as an explanation for the same observation (Kulkarni et al., 2019). While this foundational information is important for contextualizing the work, the authors should consider whether Figure 1 and the first section of the Results would be more appropriate in the Supplementary Data.

As reviewer 1 stated, this is a foundational information for contextualizing our work. In addition, this point has also been raised by reviewer 2 (reviewer 2, major point #3). Hence, we decided to keep this result and expand it to take all comments into consideration.

As mentioned, the idea that “endogenous PorP/SprF proteins produced is insufficient to ensure the efficient secretion of both the native Type B substrate and the corresponding highly produced sfGFP CTD fusion” was proposed by Kulkarni. Hence, they had to co-overproduce the SprF shuttle to detect secretion of the highly produced sfGFP-CTD(SprB) fusion. To quantify the amount of PorP/SprF produced at endogenous levels or from the replicative plasmid used in this study, we tagged the shuttle proteins (SprF and Fjoh_3951) with an ALFA tag and performed Western blots analysis to quantify the protein levels produced from the endogenous locus and from the plasmid used for overexpression. SprF-ALFA and Fjoh_3951-ALFA fusion were detected when overproduced and supported the secretion of their cognate substrate, sfGFP-CTDSprB or sfGFP-CTDFjoh_3952, respectively (new Fig. 1C, D). However, they were hardly or not detected when expressed from the endogenous locus. This result demonstrates that the low abundance of the shuttle protein is responsible for the limited interaction and secretion of the cognate CTD substrate. We modified the first section (line 107-123) of the results and Figure 1 accordingly.

Line 128: The term “linked” may not be the most accurate choice here. “Adjacent” might be more suitable, as “linked” could imply a functional or transcriptional connection, such as co-transcription as a single ORF, for which no current evidence exists.

We modified the text line 128 to take this comment into account.

Line 153: “These results suggest that Fjoh_1677 is the shuttle protein.” – This may be premature. The experiment establishes an interaction, and in this context, suggests that Fjoh_1677 may be the shuttle protein, but additional evidence is needed for a definitive conclusion.

The reviewer is correct. At this point, our results support a hypothesis and do not demonstrate that Fjoh_1677 is the shuttle protein in charge of the three orphan substrates. Hence, we were careful to use the word “suggest” in the sentence, that we would like to keep (line 156):

“These results **suggest** that Fjoh_1677 is the shuttle protein supporting the secretion of the three orphan T9SS Type B substrates Fjoh_1123, Fjoh_1720 and Fjoh_4934”.

Line 171: “These alignments also showed that, while the vast majority of sequence identity between Type B CTDs is 20-30%, other Type B CTDs share >50% sequence identity (Table 1), suggesting that cross-talks might be possible between these CTD/shuttle pairs.” – This sentence is unclear and should be rephrased to explicitly clarify which groups are being compared.

We slightly modified this part (section starting line 180). In addition, each comparison with Type B CTDs sharing more than 50% identity is now highlighted in purple in Table 1.

Line 179: “...-specific shuttles” – The listed proteins are shuttles, not cargos. This appears to be an error.(for eg: “...Fjoh_3952, or with the Fjoh_1645, Fjoh_3971 and SprB - specific shuttles” would be correct)

We corrected this error in the text (now line 194):

“The sfGFP-CTD_{Fjoh_3952} fusion was co-produced with its cognate shuttle, Fjoh_3951, or with the shuttle proteins handling Fjoh_1645, Fjoh_3971, or SprB (Fjoh_1646, Fjoh_3972, or SprF, respectively)”

Line 185: In addition to sequence similarity, the alignment of AlphaFold structures deserves mention here. A brief note on structural comparisons would add valuable insight.

This important comment was also suggested by reviewer 3. We therefore largely exploited AlphaFold structure and complex prediction. Therefore, results sections starting lines 181 and 205 have been significantly edited.

Line 192: While limiting the comparison to the last 100 amino acids of the CTD may be practical, have the authors considered that the functional Type B CTD extends to ~150-220 amino acids? Given this, it may be worth conducting multiple sequence alignments on the entire CTD region to identify additional conserved motifs.

Unfortunately, we did not detect other conserved motifs beyond the last 100 amino acids of the CTDs. We hypothesize that the function of the extended region of the CTD (upstream of the last 100 amino acids) may rely on structural determinants rather than specific amino acids.

```

Fjoh_2273 ----- 10 20 30 40 50 60 70 80 90 100
Fjoh_4538 NNAKTDPNITIDASGNVNVSSLTPAGTYTYMYRICKDKLSA----- ENCDT-AIVTITVAPNGTSMITSEACNDDTT-----
Fjoh_1645 ----- LTVQVCEKAND----- TNCST-ATLVNVEVPAIALVKTAAFNDENN-----
Fjoh_3952 ----- LTYQICEKANP----- ENCTD-AIVKIFVEPSTIVVMKVELNDHNN-----
Fjoh_1985 ----- FTICADQMF--DCSVSQVP----- D--YTKMIFNTDNTQPN-P-VIIQ--SP-----
Fjoh_3478 ----- GTVTVSWSIDIGPGRVEVS----- Y-INNCN--ENTTKFLMNVNATC-SDITITNVVD-----
Fjoh_3971 ----- ANDYLNI AQTVA S-NQFDPDPSKPNP----- DNGDQSED--DEDESEFVN--IP S-TDIAINKVEVD-----
Fjoh_4750 ----- ----- PVEVNMPE S-VVLDPKYTVSYDCVNNASNM-----
SprB ----- IY----- -----
Fjoh_1720 ----- TTYTAEANNAGCINSRRTPVHIEVYTPPVVTNE---NVI LQCS-KTV--
Fjoh_1123 ----- NRLCNSETS LDFVNDLPT-VMIEK---TYFLCNLEASL--
Fjoh_4934 ----- YARVVNGPDCYDVVPPVLVNTFDPNPFENE---SEYLCKN-DQI--

Fjoh_2273 ----- 110 120 130 140 150 160 170 180 190 200
Fjoh_4538 ----- 5D----- GQNF CGL-TNPTI-GDL----- TND
Fjoh_1645 ----- NTIANAGETISYRFTVT----- NTGNVPL-SGITI-TDLLPG-VVVSQAALD----- LNV
Fjoh_3952 ----- NDNAEAGETLTYTFTIT----- NNGNVPL-HNITI-SDLLPG-VVITGGPIS----- LGV
Fjoh_1985 ----- LPGTKFSQDMGITIKVSDKSNNISSECSF-HLFAY-PVLVDAGEDIE----- INE
Fjoh_3478 ----- NPTPNFGEHVTFTITVN----- NVGEGSF-INTIV-SEILP SGYDL-----
Fjoh_3971 ----- QDAAMGSEVIFETIAE----- NLGNLTA-TNVEV-QDILPKGYLL-----
Fjoh_4750 FVGHVKGGLPPYTLNWSVSGIVSAGN--- NEIMNTE N-NCLVIFSVTD-SFCCKAEVAYNNTFVLCGTANFSTGSGYCKDMYDLYSIYDPI TFTNLAT
SprB VVITIDPSSNPPADIDYSLDGGTIQPGNIF--TNIPAGD--HTIRVR--H-TNCTADVDFNIIIGYAPLQTL---TEEK----- GV
Fjoh_1720 ---TLDAGISGMSYLSWNG-ATTQ---TDITKAGT---YTVDVTSPPS--ENCTSRKTI VDEHYFPEINRIIIN-----
Fjoh_1123 ---HVNVAANLDSYTWKFN-DTDIVSNTYEADLINAGK---YTLTVGKIQN-DIYENKFFELVRSVLPITIKQIRYQELS----- DN
Fjoh_4934 ---TLTVAAGFSSYLSWNG-ETSNS---AII SAPGD---YSVTVK---D-ANGCEKTKNFKVI LSEPATITNAAVKDFS----- GN

Fjoh_2273 ----- 210 220 230 240 250 260 270 280 290 300
Fjoh_4538 ----- SNNTNIPATVWVWDAPDN---GNLLSASTRLTEQGRY YG---FNF---PNSACFSSEYIEVTVALTDCDVPN-DFFI PDG FSPNGDGC
Fjoh_1645 GE TRALQGNILNAHGLALGNVQFEYKITNENCP----- RSILLT----- M----- EVNDDCKVLACENI LVHNAFSPNGDGC
Fjoh_3952 GE SNDTNFSALYKIIQTIDINSKGVSNQ-ASVQG---KSARGVVVEDNSDYENIDG----- DKPPTVLDL-NGCKIKVLNAFSPNGDQ
Fjoh_1985 GE SDSHFTTGTITLTQADINAGTVVNO-ATVTG---TTQSGIKVEDKSDAANENG----- DAPTEIDV-NGCKIKIFNAISLNGDN
Fjoh_3478 GQFVKLQAVALENGS---FSWSPSAG-LNNTKVGNP IATPQETTIT---VVF-TN--KEGCQA----- EDSVTITVIPLEKDETKYGFSPNGDGC
Fjoh_3971 ---VSFNTTGGTYDPATQLWTIPALASQSLLVLI VAEVLP SGNV LNVAALIEISTPLDVAANN----- SASAS---VEPI-CLTVYNEFTPNNDG
Fjoh_4750 GDFTLIS---WDFGDNFS---NEE---NPKHIYTKVGTIT I---KQVTYTPFGCQY---SYSATIKVEKYSLLIMPNAFTPNNDG
SprB WNVITAS---AVGGGGEYVYSIDGVNFS---ETKFKIYKGTIT I---T-VRDKNCGTD---TKDYI---IEYVDVCLDNYFTPNNDG
Fjoh_1720 GTQVEIQ---LKKKEEYFEYSVDGIFNQD---SNI FYDYPGGLHTA---Y-VKEKNGCGG---IGL-D---FVVLVYPAFFTPNNDG
Fjoh_1123 ---NFIEI---IPTDGSLEYSIDGINYQN---SNYFSNIQGGTYV---Y-LRDKEGCGG---DSK-E---VTVI DYPKFTPNNDG
Fjoh_4934 DNSVLLI---EYTGTDGNYEFLSDGLTTFQD---NPLFTAVATGTIYNA---I-AKDKNGCGL---SNSFL---LVLDYPRFETPNNDG

Fjoh_2273 I NDSFV I KDIE--FLYPNYTL EFNRYGNM Y-KGDKNKP AW--DCMNYEKSGI-AGGVAPNGVYFVYLHFNKDNK-----PPKQR L Y ENR
Fjoh_4538 K NDFVFLIDGIDGLTCTPENTVEIYNRWGILVVF-ETHNYNNTTNAFDGT SRGRTTIRQSEGLPTGTIFYIVTYKSVDCGNVVIQNNKKEG L Y SK
Fjoh_1645 K NERFYIQG---LECYPENTVEIYNRWGILVVF-DVDHYNNVDRVFKGYSFGRRTTMKQSEGLP VGTIFYIILKYKSDSDSNP-----HETSGLYI INK
Fjoh_3952 M NERFYIRG---IECYPENTVEIYNRWGILVVF-ERDHYNNNDIVFKGYSFGRRTTVKESNGLPECTIYIVRYKDNNSNP-----KQEA G Y L Y I K
Fjoh_1985 I NDFWEI DK---ITDYPENELVYSRWGDLVY-OTKCYDNSTNVFSGIANKSRNL-GASQLPECTYFFEIRVNOPHH-----FKKLG L Y L K R
Fjoh_3478 A NDLFRIDC---IESYPNNE LKVFNRYGALVY-SKQHYENDW---DGTANVSGVNRGDM L PCTIFYFVITICDGT-----VK-KGWL S I MR
Fjoh_3971 Q NDTFYIDC---ITQYDNQLEIFNRWGNLVY-YKRCYDNTW---DGKADGS---AKTLRECTYFVYLDLNGNS-----AKKSGWLYI K-
Fjoh_4750 Y NDTFAFVF---LG-LSDI TLDVFDTWGCVIYTEKGTNIRGW---NGKVK---D-DIAENENY Y K I LKTFYNYHT---IVEKCAFTI I K
SprB V NDTWGPCC---TNIYNHLKFSIFDRYGRVIA-KYT-YGCKW---DGRYN---GEELPSGDYVYVLLKNDENDG---REVFCHFTYR
Fjoh_1720 Y NDLWEVTG---MENYPOAQT I FDRYKGLIA-QLNASKMSW---DGTFE---KTPMPASDYWKALKIDDSK-----PILRCHFS L K R
Fjoh_1123 Y NDFWHIKN---TSKFPNSKISIFDRYKGLIK-ELFANDHG---DGFYH---GSQMPADDYWFKANFNENI---NFSGHFS L K R
Fjoh_4934 Y NDLWVIED---SNVLPNYTIHIFDRYKGLIK-EMQNNSPGW---NGLFN---GQQLPSDDYWF T LTFADGR-----NVKGHFS L K R

```

Line 197: “B3942” – This appears to be a typo.

Corrected.

Line 229: “86% of the cells” – A typo?

This is not a typo. Here we looked at foci subpopulations. As some cells present multiple foci, both static and highly dynamic behaviors can be observed in a single cell. Actually, 42% of the cells present both static and highly mobile foci.

Line 230: How exactly does the motion of RemA-Halo differ from RemA-CTD? The figure legend also does not describe the motion. A brief statement clarifying whether the motion is helical or follows another pattern would be useful. Additionally, providing a link to a supplementary video would enhance visualization for the reader. Since RemA was found to move helically on the cell surface similarly to SprB (Shrivastava et al., 2012), and the findings here differ, this discrepancy should be discussed.

To help visualization, we added supplementary movie files for each Halotag fusion shown (Movie S1 to S9). The rainbow trace panels in Fig. 6 and 7 highlight the helical movement of relevant fusions.

Concerning RemA, we did not detect any helical behavior with our RemA-Halotag fusion. However, we detected secretion of RemA-Halotag in the supernatant. Hence, our fusion may be only partially

functional for secretion compared to the myc-RemA fusion that Shrivastava and colleagues observed in 2012. We discussed this discrepancy line 254:

“It is noteworthy that we did not detect any helical behaviour of RemA-HaloTag, although RemA was found to move helically in a previous study⁴³. This may be explained by the use of different tags, inserted at different positions, to track the protein.”

Figures 6 and 7: The number of cells observed should be stated in the legend, even though it is mentioned in the main text.

We added the number of cells observed in the figure legend.

Line 253: “By contrast, the Type B CTD of SprB addresses any substrate to the gliding machinery.” – A citation should be added here.

Based on our result, we meant to write (now line 280):

“By contrast, the Type B CTD of SprB seems to address any substrate to the gliding machinery”

Line 357: Precipitation of extracellular proteins – What volume of culture was taken? This is crucial information for reproducibility and is missing in other experiments as well, such as in the Cross-linking and co-immunoprecipitation.

We added this information in the corresponding paragraphs of the Experimental Procedures section.

Line 368: Western Blot and Dot Blot Analyses – Missing details: How much protein was loaded per well? What was the starting volume of culture for protein harvesting? These details should be included for reproducibility.

We added this information in the corresponding paragraphs of the Experimental Procedures section.

Line 396: Sonication parameters should be specified, as they are crucial determinants of the experiment's outcome. Additionally, DDM is mentioned without its full form—this should be corrected.

We added this information in the corresponding paragraph of the Experimental Procedures.

Figure 1:

The schematic should specify which protein's CTD is being tested in this experiment or at least indicate that it is a CTD-B. It would also be helpful to mention the presence of a Sec signal peptide at the N-terminal end, especially since it is referenced in panel B. The legend states: SprB and Fjoh_3952, with and without coexpression of SprF (SprB-SprF) and Fjoh_3951 (3952-3951) respectively,” but the textual annotation above the gel does not clearly indicate where the “without SprF” data is shown, if shown at all. The formatting of this panel may

need adjustment to make it clearer which lanes correspond to CTD-B and whether a cognate shuttle is co-expressed. Ensuring this will make the figure more technically accurate and accessible, especially for junior researchers and those from outside the field. Additionally, the legend should explicitly mention that “SP” refers to “signal peptide.” Also, where sfGFP_{peri} is mentioned, it is technically incorrect since this construct does not contain any CTD.

We modified Figure 1 to specify the CTDs being tested. We also modified Figure 1 to include all comments from reviewers 1 and 2. In particular, there is now a schematic representation on the chimeric sfGFP-CTDs-shuttle constructs tested. Finally, Supplementary figure 7 gathers all the constructs tested for secretion in this study.

Figure 2:

The letters following “CTD” should be subscripted for clarity and consistency.

- The base of the logarithm used should be specified in the figure, as the Methods section mentions a base of 2—this is worth clarifying to avoid confusion.

Corrected.

The values used to generate Figure 2 were converted back to linear scale (they are no longer in $\log(2)$), in order to present absolute LFQ ratios more intuitively.

In the figure, we used a logarithmic scale for the y-axis to better visualize the large range of the data. To avoid any misunderstanding, we removed the mention of the logarithmic scale in the figure.

• The legend states:

“When the protein was absent from the sfGFP_{peri} strain, an arbitrary value was assigned to calculate a ratio, and the bar has been colored grey.”

The Methods section should further elaborate on how this computation was performed. Specifically:

- How was this “arbitrary value” chosen?
- Does it vary across different CTDs, such as in subpanels E and F?
- Does the choice of this arbitrary value affect the ranking of the candidate protein (F_{joh_1677}) on the x-axis?
- Is using an arbitrary value a standard approach in this type of analysis, or could a fixed value, such as “1” or its equivalent in log-transformed form, be used instead for normalization?

Clarifying these points would strengthen the methodological transparency of the analysis.

We modified the Experimental procedures section to take into account all these remarks.

Using arbitrary value is a common approach in this type of analysis to calculate LFQ ratios when a protein is absent from the negative control (Lazar et al., 2016). This lack of detection is generally attributed to protein levels falling below the limit of detection rather than a complete absence. Missing values were replaced using data imputation by randomly selecting from a normal distribution centered on the lower edge of the intensity values that simulates signals of low abundant proteins using default parameters (a downshift of 1.8 standard deviation and a width of 0.3 of the original distribution). In the original manuscript, we used a fixed value. In this revised version, we used this imputation method to limit bias in the analysis. We modified the Experimental procedures section accordingly (line 484):

“Missing values were replaced using data imputation by randomly selecting from a normal distribution centered on the lower edge of the intensity values that simulates signals of low abundant proteins using default parameters (a downshift of 1.8 standard deviation and a width of 0.3 of the original distribution).”

Figure 3:

The authors should consider consolidating two panels into one, grouping data as CTD-without-shuttle – CTD-with-shuttle pairs (e.g., 1123 | 1123+1677 , 1720 | 1720+1677 ...). This would significantly improve figure readability.

If the current layout is retained, Panel A should also include a schematic similar to Panel B, effectively recapitulating part of Figure 1 to provide better context for the reader.

We agree that this would improve figure readability. We have modified the figure as suggested by the reviewer.

Figure 4:

The “null set” symbol should be replaced with the stain annotation and the figure legend should describe the subsets.

Corrected.

Reviewer #2 (Remarks to the Author):

Main comment:

The authors have performed an interesting study regarding the Type B CTD/ shuttle protein interaction and secretion in Type IX secretion system. Multidisciplinary methodologies are combined to tackle fundamental questions and draw interesting conclusions/hypothesis, expanding our knowledge in the field of type IX.

However, the quality of the research is not well translated in the manuscript and several key points need significant improvement regarding the clarity, organization and data presentation within the manuscript. Description of the results and figures are too elusive; the manuscript is written in a rush and not well structured leading to a not concise presentation of the main arguments. After careful consideration, I am unable to recommend this paper for immediate publication, unless the comments below are addressed and the new manuscript be reviewed again.

Major comments:

1: the manuscript is poorly written, and consequently lacks of consistency. As a reader, I feel like different sections have been written by different people and simply merged together (see lines 125-156, written in a completely different style than the rest of the manuscript). A lot of grammar, typos, spelling mistakes are present (some are listed below); and the phrasing used by the authors is misleading since the interpretation is not clearly connected to the presented data. Additionally, experimental procedures are poorly described and table legends are missing. As a reviewer, I suggest that the authors seek the assistance of a native English speaker to improve the overall quality of the manuscript.

Some examples of inconsistency across the manuscript:

a) Table S1 and S2 legends are missing, and although are called S1 and S2 in the main text, in the supplementary files they are called “strains”, “Plasmids”.

b) The quality of the figure seems to be below the requirement of the journal (the resolution and formatting need revision). This might be an artefact of the generated files of course.

c) Line 97, 366 and 368: should be “analysis” instead of “analyses”

d) line 405: should be “7 min” instead of “7 mn”

Many more examples could be provided; the manuscript need to be properly edited.

We truly apologize if the reviewer felt the manuscript was badly written. In this revised version, we edited the text to significantly improve writing, with the assistance of a native English speaker. We corrected grammar, typos, spelling mistakes. Experimental procedures have also been updated, and details added to the figure legend.

2: The names of the secretory proteins and protein derivatives used in the study are long and complicated, confusing the reader and breaking the flow of the paper. The authors could include a schematic representation of the system including proteins and engineered derivatives. It could be either in the main figures, or added in the supplementary materials. A concluding cartoon/ model showing chaperones binding (or not) to secretory proteins would also be an additional value for the paper and very helpful for a non-expert reader to comprehend all the results described in the present study.

In Figure 1, we included a schematic representation of the chimeric sfGFP-CTD-shuttle constructs tested. Figure 1 and Figure 3 are also accompanied by Supplementary Figure 7, in which all the sfGFP-CTD constructs tested in this study are listed. In addition, we added a schematic model showing CTD/shuttle specificities in a new figure (Figure 8).

3: Lines 115-123, 160-168, Fig. 1 and 3: The authors discuss about limiting interactions between chaperones and secretory proteins due to limited amount of proteins produced intracellularly. However, there are no data demonstrating this hypothesis. Since the authors are growing bacteria and collect samples for their secretion assays, they should perform Western blots analysis to quantify the amount of those proteins produced inside the bacteria. The authors might then demonstrate that the low protein abundance is indeed responsible of the limited interactions between chaperones and secretory proteins, resulting in a defective secretion phenotype. Since this is one of the main statement of the paper controls experiments of gene expression levels are essential to convince the reader.

This major comment is in line with the comments of reviewer 1 on Figure 1 and reviewer 3. We have conducted additional experiments to answer the criticism.

Here is the answer from above:

As mentioned, the idea that “endogenous PorP/SprF proteins produced is insufficient to ensure the efficient secretion of both the native Type B substrate and the corresponding highly produced sfGFP CTD fusion” was proposed by Kulkarni. Hence, they had to co-overproduce the SprF shuttle to detect secretion of the highly produced sfGFP-CTD(SprB) fusion. To quantify the amount of PorP/SprF produced at endogenous levels or from the replicative plasmid used in this study, we tagged the shuttle proteins (SprF and Fjoh_3951) with an ALFA tag and performed Western blots analysis to quantify the protein levels produced from the endogenous locus and from the plasmid used for overexpression. SprF-ALFA and Fjoh_3951-ALFA fusion were detected when overproduced and supported the secretion of their cognate substrate, sfGFP-CTDSprB or sfGFP-CTDFjoh_3952, respectively (new Fig. 1C, D). However, they were hardly or not detected when expressed from the endogenous locus. This result demonstrates that the low abundance of the shuttle protein is responsible for the limited interaction and secretion of the cognate CTD substrate. We modified the first section (lines 96-123) of the results and Figure 1 accordingly.

#4: The number of biological/ technical repeats for each experiment needs to be mentioned. Moreover, the statistical significance of the comparison of the MS data should be calculated and included in Fig. 2.

We added these details in the Experimental Procedures section and in the Figure legend.

#5: In Fig. 2C, SprB fusion protein is missing, so I assume that the protein was not detected by MS. Indeed, protein identification using LC-MS, especially after in-gel digestion, could lead to no protein

detection for various reasons (poor peptide fractionation, low abundance, effect of ionization, or not stably produced protein). The authors need to consolidate their hypothesis, rule out experimental/technical limitations (the production of proteins needs to be demonstrated by gel analysis) before interpreting their data and mention it in the main text.

Indeed, in the case of the Co-IP with sfGFP-CTD_{SprB}, SprB did not show in the top LFQ intensity ratios. This may be explained by the level of production of endogenous SprB (as observed by the presence of numerous filaments attached to the cell surface; Liu et al., 2007 *J. Bact*; Nakane et al. 2013 *Proc Natl Acad Sci U S A*) and the very large size of the protein (6473 amino acids after cleavage of the predicted Sec-dependent signal peptide). Thus, SprB was also significantly detected in the negative control strain (sfGFP_{peri}), resulting in a lower LFQ intensity ratio for SprB. We added the LFQ intensity ratio found for SprB in Fig. 2C and inserted two sentences in the text to take this comment into consideration (line 151):

“However, we noticed that SprB was not among the highest LFQ intensity ratios (Fig. 2C). This is because SprB is a very large protein that was detected in significant amount in the control experiment.”

6: “Material and methods” section needs extensive revisions as critical technical details are lacking. As an example, the authors are complementing bacteria in trans with plasmids carrying genes for secretion and co-purification assays. However, there are no details on the gene expression conditions (inducer concentration, time, temperature ...). Again, I am wondering how the authors secure that protein over-production (higher than chromosomal levels I suppose) does not lead to un-physiologically relevant interactions: a chaperone might interact with a secretory protein only because the production level is so high that unspecific interactions are showing, or because they are co-produced. This should be discussed in both the results and discussion sections.

We included the details requested by the reviewers in the Experimental Procedures section. In particular, we added quantitative information when it was missing.

The P_{remA} promoter used to overexpress the different constructs has been used in previous studies with the same purpose (Kulkarni et al, 2017 *Journal of Bacteriology*, Kulkarni et al, 2019 *Journal of Bacteriology*). Although there is no specific study on this promoter, it has been observed that it is highly active in exponential phase and early stationary phase of growth.

We cannot strictly rule out the hypothesis that over-production leads to un-physiological interactions. However, as already observed by Kulkarni and colleagues, the interactions observed between over-produced cognate pairs of secretory protein (CTD) and shuttle protein appear highly specific.

7#: The authors defined 5 conserved motifs after sequence alignments between the Type B CTDs. However, it is not clear why a protein would need all 5 motifs for shuttle recognition and secretion, if swapping just one is enough to abolish one or both functions. One should swap all or more motifs to shed more light to the interaction mechanism and provide insight on why those motifs are / are not necessary for efficient secretion. The authors should do those experiments to strengthen their current results and conclusions and provide additional value to the present manuscript. Moreover, the reason why they focused only on motif B and E is not very clear. The authors mentioned “notable differences between these motifs” (line 193), however, they should elaborate on this statement more and justify their choice.

We have redesigned this section (starting line 204) of the manuscript for clarity.

In the revised manuscript, we have now included results of the mutation of each of the motifs (A to E), as well as the swapping of all 5 motifs.

First, we asked if each conserved motif is necessary for secretion and/or specificity. To address this question, we used AlphaFold3 to predict the structure of the CTD of SprB (Fig. S5A). For each

motif, we then designed point mutations that do not impair the global structure, as predicted by AlphaFold3 (Fig. S5B, C). We then tested the effect of each motif mutation on secretion. Each motif mutant was not secreted (Fig. 5B), indicating that each conserved motif, A to E, is required for secretion.

Second, we asked if swapping all five motifs (A to E) of CTD_{SprB} to those present in CTD_{Fjoh_3952} would be sufficient to allow the recognition of CTD_{SprB} by the Fjoh_3951 shuttle (that normally recognizes CTD_{Fjoh_3952}). Our new results (Fig 5D) show that this new construct is not secreted in the presence of SprF nor in the presence of Fjoh_3951, suggesting that the 5 conserved motifs are not sufficient to recapitulate specificity toward a shuttle protein.

Our data support the idea that each motif, A to E, is required for T9-dependent secretion, but all five motifs participate but are not sufficient to achieve specific recognition by a shuttle protein.

Minor comments:

1: Figures resolution and formatting is poor in the printed version. Authors should check if the figures meet the journal requirements

We checked that the original figures meet the journal requirements. We suspect that file conversion during submission was the problem.

#2: In the last paragraph of the results (lines 202-274), the authors mentioned “CTD dictates protein “dynamics”. However, from the data shown (Fig. 6 and 7), one could suggest that CTD dictates protein localization and function. Protein dynamics as a term refers to protein domain/chain movements, structural alterations and conformational changes. This study does not contain data demonstrating changes in dynamics (i.e structural analysis, MD simulation). Therefore, the authors should be carefully and justify the use of “dynamics” or replace it with the word “function” or activity that seem more proper according to their data.

We acknowledge that the word “dynamics” may be confusing here. We have replaced dynamics at different places in this paragraph:

“This raises the idea that each shuttle serves not only to collect the substrate at the exit of the translocon but also to transfer it to its final **localization after secretion**.”

“The second subpopulation corresponds to highly mobile RemA-HaloTag foci (observed in 86% of the cells presenting fluorescent signal, $n=86$) (Fig. 6C, *middle panel*, Supplementary Movie 3), with a **behaviour** different from that of the SprB adhesin.”

#3: In Text S1, the cloning description is very complicated. A table listing all the suicide plasmids generated in this study (mentioning the name of the plasmid, the PCR product/insert generated, the template used and the primers used) would consolidate the procedure and improve the presented manuscript.

We have now included Supplementary Table 5, listing all the suicide plasmids generated in this study, including the name of each plasmid, the corresponding PCR products/inserts and the primers used. As the template DNA was predominantly *F. johnsoniae* genomic DNA, we did not include a dedicated column for it in the table. However, in cases where a different template was used, this information is specified in the table legend.

4: line 11: keywords are missing.

Added

#5: line 91-96: A reference is missing from this statement.

Added

#6: line 211: consider changing “collect the substrate..” with target or escort or transfer

We believe the reviewer meant to replace the end of the sentence (now line 234):

“This raises the idea that each shuttle serves not only to collect the substrate at the exit of the translocon but also to **transfer it to its final localization after secretion.**”

Reviewer #3 (Remarks to the Author):

In this manuscript, Paillat et al. have studied how substrates of the Type IX Secretion System are recognised by shuttle proteins following secretion, which facilitates correct localisation of these substrates. C-terminal domains (CTD) on these substrates are recognised by these shuttle proteins and while Type A CTD are well-characterised, less is known about how Type B CTD are involved in substrate localisation. It has previously been shown that Type B CTD are encoded upstream of their cognate shuttle protein, for which they are specific.

Using co-immunoprecipitation mass spectrometry, a single shuttle protein responsible for the localisation of three orphan Type B CTD proteins was identified. These had higher homology in the CTD compared to other Type B CTD substrates, which suggested a mechanism by which these three substrates could be recognised by the same shuttle protein. Sequence conservation analysis identified five motifs that were conserved in all Type B CTD found in this strain, and mutagenesis of some of these demonstrated they were essential for secretion. Finally, it was shown that the CTD alone determined the extracellular localisation of these substrates, being targeted to the gliding motility apparatus, cell surface anchored or released from the cell. Altogether this work presents some interesting findings in regard to T9SS substrate targeting, however I have a few concerns which are detailed below.

Major comments

Figure 1: It is unclear to me why Fig.1 has been included in the results. From what I can tell, the first part of panel B has been published in Kulkarni et al., 2017 and the second part has been published in Kulkarni et al., 2019 and so, should be included in the introduction. If a novel finding is to be presented from this experiment, it should be the focus of the discussion in text. Currently, lines 113-115, discuss the reliance of Type B CTD for downstream encoded SprF-like proteins, which cannot be concluded from Fig. 1 as a negative control is absent. In addition, it is not clear what panel A is meant to convey to the reader, given that it has little relation to the proteins presented in panel B.

This comment is related to comments of reviewer 1 and 2. As mentioned previously, in response to reviewer 1 and 2, we have modified Fig. 1 to take into account all reviewers criticisms and suggestions. Fig. 1 has been expanded with Western blot data showing production levels of SprF and Fjoh_3951 shuttle proteins from the endogenous loci and from the replicative plasmids used.

Copied from above:

As mentioned, the idea that “endogenous PorP/SprF proteins produced is insufficient to ensure the efficient secretion of both the native Type B substrate and the corresponding highly produced sfGFP CTD fusion” was proposed by Kulkarni. Hence, they had to co-overproduce the SprF shuttle to detect secretion of the highly produced sfGFP-CTD(SprB) fusion. To quantify the amount of PorP/SprF produced at endogenous levels or from the replicative plasmid used in this study, we

tagged the shuttle proteins (SprF and Fjoh_3951) with an ALFA tag and performed Western blots analysis to quantify the protein levels produced from the endogenous locus and from the plasmid used for overexpression. SprF-ALFA and Fjoh_3951-ALFA fusion were detected when overproduced and supported the secretion of their cognate substrate, sfGFP-CTDSprB or sfGFP-CTDFjoh_3952, respectively (new Fig. 1C, D). However, they were hardly or not detected when expressed from the endogenous locus. This result demonstrates that the low abundance of the shuttle protein is responsible for the limited interaction and secretion of the cognate CTD substrate. We modified the first section (line 107-123) of the results and Figure 1 accordingly.

Line 186/299; Fig. 5: While the motif analysis in Fig. 5A is compelling, the subsequent experiments do not really address the importance of these motifs. Can the authors address why motifs A, C and D were not tested?

In addition, it is not clear which sequence in the alignment represents SprB, or if it is present at all. It is therefore not clear what residues have actually been mutated by switching the B and E motifs. In addition, if the reason that only B and E were tested is because A, C, and D were identical between SprB and Fjoh_3952, then this would suggest that regions outside the conserved motifs are also important for shuttle specificity. Further clarity and evidence is required to support the importance of these Type B CTD motifs in recognition by shuttle proteins. Would it be possible to use AlphaFold to model the CTD with the cognate shuttle protein to give insight into what residues may be facilitating recognition of the CTD? This could then be used to either support why B and E are important in the process and give insight into what other residues in the CTD are involved.

This criticism was also raised by reviewer 2. We significantly redesigned this section of the results (lines 204-229) to include suggestions from both reviewers.

Copied from above:

In the revised manuscript, we have now included results of the mutation of each of the motifs (A to E), as well as the swapping of all 5 motifs.

First, we asked if each conserved motif is necessary for secretion and/or specificity. To address this question, we used AlphaFold3 to predict the structure of the CTD of SprB (Fig. S5A). For each motif, we then designed point mutations that do not impair the global structure, as predicted by AlphaFold3 (Fig. S5B, C). We then tested the effect of each motif mutation on secretion. Each motif mutant was not secreted (Fig. 5B), indicating that each conserved motif, A to E, is required for secretion.

Second, we asked if swapping all five motifs (A to E) of CTD_{SprB} to those present in CTD_{Fjoh_3952} would be sufficient to allow the recognition of CTD_{SprB} by the Fjoh_3951 shuttle (that normally recognizes CTD_{Fjoh_3952}). Our new results (Fig 5D) show that this new construct is not secreted in the presence of SprF nor in the presence of Fjoh_3951, suggesting that the 5 conserved motifs are not sufficient to recapitulate specificity toward a shuttle protein.

Our data support the idea that each motif, A to E, is required for T9-dependent secretion, but all five motifs participate but are not sufficient to achieve specific recognition by a shuttle protein.

Minor comments:

Figure 2: In most samples tested, the expressed fusion protein is detected as one of the most abundant species, however in Fig. 2C, SprB does not appear to be detected. Is it known why SprB cannot be detected?

This point was also raised by reviewer 2.

Copied from above:

Indeed, in the case of the Co-IP with sfGFP-CTD_{SprB}, SprB did not show in the top LFQ intensity ratios. This may be explained by the level of production of endogenous SprB (as observed by the presence of numerous filaments attached to the cell surface; Liu et al., 2007 *J. Bact*; Nakane et al.

2013 *Proc Natl Acad Sci U S A*) and the very large size of the protein (6473 amino acids after cleavage of the predicted Sec-dependent signal peptide). Thus, SprB was also significantly detected in the negative control strain (sfGFP_{peri}), resulting in a lower LFQ intensity ratio for SprB. We added the LFQ intensity ratio found for SprB in Fig. 2C and inserted two sentences in the text to take this comment into consideration (line 151):

“However, we noticed that SprB was not among the highest LFQ intensity ratios (Fig. 2C). This is because SprB is a very large protein that was detected in significant amount in the control experiment.”

Figure 3: The presence of the gene diagram is beneficial to the reader for clarity, however, should be included in both panel A and B, not B alone. This is also true for Fig. 1 where it is only beneficial for the reader if it provides a visual reference for what is present in the figures.

For more clarity, we updated Fig. 1 and Fig. 3, with schematics of the constructs tested. We also added Supplementary Fig. 7, in which we gathered all the constructs used in Fig. 1 and 3.

Figure 6-7: There are several panels missing from both Figures 6D and 7. Is the data not available or is there a reason it has been excluded?

In Fig. 6, we did not include kymograph when there was no signal. In this case we put N/A (for Non applicable). In Fig.7 we added the missing rainbow trace panels.

Figure 5: As mentioned above, if SprB is present in the alignment in panel A, can it be labelled so it is easier for the reader to identify. Also, in the legend for 5B and C, it implies the BE double mutation was also tested for SprF, which doesn't seem to be true.

Fig. 5 has been largely modified. The comment has been addressed in the revised figure.

Spelling and grammar:

Line 156 – “Cross-talks” should be cross-talk. Corrected

Line 181 – “Fjoh_3971 and SprF” should be ‘or’. Corrected

Line 221/257 – “upstream the...” should be ‘upstream of the...’ Corrected

Line 253 – I don't think addresses is the correct word here. Perhaps ‘directs’ or ‘targets’
We chose to use the word “targets” instead of addresses. Accordingly, we modified the manuscript title to:

“Specialized shuttle proteins recognize Type IX secretion signals and **target** effectors to their final destinations”

Reviewer #4 (Remarks to the Author):

The authors primarily investigated the substrate recognition and secretion mechanisms of the Type IX Secretion System (T9SS) in bacteria, particularly members of the Bacteroidetes phylum, with a focus on the function of Type B C-terminal domains (CTDs) and their dedicated shuttle proteins. In *Flavobacterium johnsoniae*, the authors discovered that among the 12 Type B substrates, three "orphan" substrates (Fjoh_1123, Fjoh_1720, and Fjoh_4934) were not genetically linked to known PorP/SprF-like genes. Through co-immunoprecipitation and mass spectrometry analysis, it was found that these three substrates share the same orphan shuttle protein, Fjoh_1677, suggesting that a single shuttle protein can recognize multiple substrates. By performing motif-swapping experiments, the authors identified that two conserved motifs (B and E) are necessary for the recognition of substrates by their specific shuttle proteins. Additionally, it was found that the CTD is sufficient to guide heterologous proteins (e.g., sfGFP) through T9SS secretion and determines the final localization of the substrates (e.g., dynamic adhesion to the gliding motility machinery or static anchoring to the cell surface). This study reveals the specific interaction mechanism between Type B CTDs and their dedicated shuttle proteins through conserved motifs, expanding our understanding of the diversity of bacterial secretion systems. The findings demonstrate that CTDs not only serve as secretion signals but also encode localization information, potentially regulating the functional division of substrates (e.g., motility, adhesion, or enzyme secretion) through different shuttle proteins. This provides a theoretical foundation for the engineering of bacterial secretion systems, such as the targeted delivery of proteins.

Major revision needs to be made:

1. In this study, co-immunoprecipitation (co-IP) and mass spectrometry (MS) were used to identify the shuttle protein responsible for the secretion of T9SS Type B CTD effector protein. But is there an independent experimental method (such as protein-protein interaction verification experiment, such as two-hybrid or surface plasmon resonance) to further confirm the specificity of these interactions?

The reviewer raises an important point.

First, to support the results from co-immunoprecipitation experiments, we used AlphaFold3 to 1) determine whether the CTD-shuttle complexes were well predicted and 2) to see whether AlphaFold3 could specify the CTD-shuttle pairs. In Fig. S3, we gathered the prediction of the five complexes predicted, all of them with an iPTM above 0.8, indicating all predictions are highly confident. Interestingly, when we injected up to 5 CTDs and their cognate shuttle protein, AlphaFold retrieved all pairs correctly, supporting the specificity between CTDs and shuttles. In addition, AlphaFold3 always associated the three "orphans" Type B CTD substrates Fjoh_1123, Fjoh_1720 and Fjoh_4934, with the Fjoh1677 shuttle protein, which is consistent with what was observed by co-immunoprecipitation.

We have added these observations in the second result section, line 160, as follows:

“Interestingly, AlphaFold3 complex predictions were consistent with the cognate CTD/shuttle pairs identified by co-immunoprecipitations. First, all cognate pairs were predicted with high confidence with iPTM scores above 0.8 (Supplementary Fig. 3A, B). Second, when multiple CTDs and multiple shuttle sequences were provided (up to 5), AlphaFold3 provided the exact same pairs we identified using biochemistry (Supplementary Fig. 3C). Finally, when multiple shuttle sequences were provided, AlphaFold3 always associated any of the three “orphans” Type B CTD substrates Fjoh_1123, Fjoh_1720 and Fjoh_4934, with the Fjoh_1677 shuttle protein, which is consistent with our co-immunoprecipitation results.”

Second, as an independent wet method, we did try pull-down experiments after co-production in the heterologous host *Escherichia coli*. We considered two CTD/shuttle pairs: SprB/SprF and Fjoh_3952/Fjoh_3951. CTDs were tagged with the VSVG or ALFA peptides and SprF and Fjoh_3951 were his-tagged. We expressed these fusions heterologously in *Escherichia coli* to perform pull-down experiments after detergent solubilization, and in the presence or absence of crosslinker. The proteins were produced and easily detected by Western blot, but we did not detect interactions. Since membrane protein solubilization may perturb these interactions, we also tried

to mix enriched tagged CTDs with intact cells expressing SprF-His or Fjoh_3951-His, in the presence or absence of crosslinker, prior to cell lysis. This approach also failed. We hypothesize that interaction may have to first take place while the CTD exits the SprA translocon. Biochemical experiments with outer membrane proteins are often very difficult!

Minor revisions need to be made:

1. Figure 1-3: The molecular weight markers in the Western blot diagram are not marked with specific values (only the kDa range is displayed), and some bands are blurred (as shown in the CE and Sup bands of Fjoh_4934 in Figure 3A). Need to provide higher resolution images and clear marks.

We added Table S8 to indicate the predicted molecular weight of each fusion tested. This Table is referenced in the figure legend.

2. The format of some references is not uniform, so it needs to be uniform.

Corrected

3. Mass spectrometry data was submitted to ProteomeXchange, but the login number was not provided (PXDxxxx is a placeholder). The actual identifier needs to be supplemented.

The mass spectrometry proteomics data have been deposited to the ProteomeXchange Consortium via the PRIDE partner repository with the dataset identifier PXD065655.

Reviewers may access the data on the PRIDE website using the following details:

https://www.ebi.ac.uk/pride/

Project accession: PXD065655

Token: YJ4hIBUhXOLZ

Alternatively, reviewer can access the dataset by logging in to the PRIDE website using the following account details:

Username: reviewer_pxd065655@ebi.ac.uk

Password: gfAANGhO2KEj

4. In line 32 change plays a crucial role for "to" "plays a crucial role in". Corrected

5. Type B CTDs can be divided into dynamic and static types. What is the biological significance of this functional differentiation? This could be discussed in discussion.

We now discuss the idea that dynamic and static Type B CTDs have different functions in the last paragraph of the discussion of the revised manuscript. In particular here (line 345):

“The CTDs of SprB, Fjoh_4750, and Fjoh_1123 direct the substrates to the gliding machinery, allowing their helical displacement at the cell surface (Fig. 8). On the contrary, the CTD of Fjoh_3952 serves to anchor the substrate to the cell surface. This suggests that the CTD of Fjoh_3952, and possibly other similar Type B CTDs, are involved in attaching adhesins implicated in cell-cell or cell-surface adhesion rather than gliding. Both these functions are likely conserved in other bacteria using the T9SS for protein delivery.”

6. It is emphasized that Type B CTD is bifunctional (secretion+targeting), and more experimental data should be provided to support this view.

Rather than “bifunctional”, we meant to convey the idea that Type B CTDs contain both the signals for T9-dependent secretion, translocation through the SprA translocon in the outer membrane, as well as signals for their final localization at the cell surface. Hence, we did not use “bifunctional”.

7. How to prove that CTD domain not only participates in secretion, but also determines the final location of protein?

This very interesting point was investigated in the last result section associated with Fig. 6. By exchanging CTDs between substrates, we were able to switch the behavior of the substrates, either rendering them dynamic at the cell surface using the CTD of SprB, or making them static using the CTD of Fjoh_3952.

8. It is suggested that the specific sequence of CTD determines the protein targeting, but can this be directly proved by deleting or replacing these sequences (such as motifs B and E)?

Indeed, we have changed the CTD sequence of SprB by the one of RemA or AmyB and showed that in these cases SprB was no more targeted to the gliding machinery but was either released into the medium or statically anchored at the surface. This result supports the idea that the targeting signal is encoded in the CTD and not in the adhesin domain of SprB.

9. Will the low level expression of Fjoh_1123-HaloTag-CTD have a certain impact?

In our experimental conditions, Fjoh_1123-HaloTag-CTD was indeed not detected. We therefore had to study the function of its CTD by generating a SprB-HaloTag-CTD_{Fjoh_1123} chimera, supporting the idea that Fjoh_1123 was addressed to the gliding machinery and dynamic at the cell surface. It will be interesting to determine the conditions in which Fjoh_1123 may be produced at higher levels to understand its function. We hypothesize that this could be related to the substratum on which cells are gliding.